# Do Counterfactually Fair Image Classifiers Satisfy Group Fairness? – A Theoretical and Empirical Study

**Sangwon Jung**[1*]  **Sumin Yu**[1*]  **Sanghyuk Chun**[2†]  **Taesup Moon**[1,3†]

[1] Department of Electrical and Computer Engineering, Seoul National University
[2] NAVER AI Lab    [3] ASRI/INMC/IPAI/AIIS, Seoul National University

## Abstract

The notion of algorithmic fairness has been actively explored from various aspects of fairness, such as counterfactual fairness (CF) and group fairness (GF). However, the exact relationship between CF and GF remains to be unclear, especially in image classification tasks; the reason is because we often cannot collect counterfactual samples regarding a sensitive attribute, essential for evaluating CF, from the existing images (*e.g.*, a photo of the same person but with different secondary sex characteristics). In this paper, we construct new image datasets for evaluating CF by using a high-quality image editing method and carefully labeling with human annotators. Our datasets, CelebA-CF and LFW-CF, build upon the popular image GF benchmarks; hence, we can evaluate CF and GF simultaneously. We empirically observe that CF does not imply GF in image classification, whereas previous studies on tabular datasets observed the opposite. We theoretically show that it could be due to the existence of a latent attribute $G$ that is correlated with, but not caused by, the sensitive attribute (*e.g.*, secondary sex characteristics are highly correlated with hair length). From this observation, we propose a simple baseline, Counterfactual Knowledge Distillation (CKD), to mitigate such correlation with the sensitive attributes. Extensive experimental results on CelebA-CF and LFW-CF demonstrate that CF-achieving models satisfy GF if we successfully reduce the reliance on $G$ (*e.g.*, using CKD).

## 1  Introduction

As machine learning algorithms are deployed in societal computer vision applications such as facial recognition [39] and job interview [29], concerns have grown regarding their potential to discriminate against certain individuals and groups. For instance, a face recognition system might exhibit disparate accuracies across different demographic groups [3], while a job interview algorithm could be biased based on protective attributes even for the same interviewee [11]. Consequently, *algorithmic fairness* in image classifiers has gained significant attention in academic and industrial research communities.

While conceptually apparent, determining a concrete notion of fairness is challenging, leading to the proposal of several different fairness notions. One prevalent notion is *counterfactual fairness* (CF) [23] which seeks consistent predictions when only a sensitive attribute is intervened. Another important notion is *group fairness* (GF) [43] that aims to treat different demographic groups equally to prevent one group unfairly disadvantaged compared to another. Many researchers have focused on developing separate algorithms to achieve each notion, while understanding the exact relationship between CF and GF is yet under-explored; *e.g.,* some recent work [1, 35] showed that a model achieving CF can meet several GF notions *only* under specific conditions of Structural Causal Models.

---

[*]Equal contribution.
[†]Co-corresponding author.

38th Conference on Neural Information Processing Systems (NeurIPS 2024) Track on Datasets and Benchmarks.

Furthermore, previous studies on the relationship between CF and GF have not considered the setting of image classification due to the absence of *evaluation* datasets with counterfactual images, in which only the sensitive attribute is altered from the original images while other attributes not caused by the sensitive attribute remain unchanged — a data nearly impossible to collect in the real world. There have been several works generating counterfactual images using generative models [4, 19, 26, 32, 44, 34, 5], but they have only focused on utilizing generated counterfactual samples for training rather than evaluation. Moreover, these methods often suffer from low-quality counterfactual images generated based on VAE [21] or GAN [9]. One notable exception is Liang et al. [24], which offers an evaluation dataset including counterfactual images. However, their images are all synthetic; thus, it is still insufficient to evaluate CF due to distribution shifts from real-world images.

In this paper, we construct CF benchmarks for image classification tasks using high-performing diffusion model-based generative models. Our datasets build upon popular facial benchmark datasets used for evaluating GF, CelebA and LFW, by altering the sensitive attribute with pre-trained Instruct-Pix2Pix (IP2P) [2]. We then carefully curate the edited samples by human annotators and verify the reliability of our datasets as counterfactual samples from additional annotators. Note that our datasets, CelebA-Counterfactual Face (CelebA-CF) and LFW-Counterfactual Face (LFW-CF), share the same test samples as the original GF benchmarks, enabling the evaluation of both GF and CF.

Using our datasets, we conduct a primitive study on the relationship between CF and GF in image classification, *e.g.*, test whether CF implies GF for image classifiers. To that end, we train CF-aware methods [36, 7] and evaluate them with our datasets using both CF and GF metrics. From the result, we observe that they achieve CF but fail to satisfy GF, contrary to previous findings that CF can imply GF [1, 35]. We attribute this failure to Structural Causal Models (SCMs) of image generation. Specifically, for an image SCM, a latent attribute $G$ is more likely to exist, which could be correlated with, but not caused by, the sensitive attributes. For example, in the case where the sensitive attribute is the sex of a person in an image, secondary sex characteristics such as beard and hairline are highly correlated with hair length, but it does not mean that such characteristics cause the length of hair. In this scenario, if a model achieving CF relies on the attribute $G$ (*e.g.*, hair length) on its prediction, it could more severely violate GF in the worst case. Therefore, if we can reduce the dependency on $G$ of a CF-aware model, we may achieve both CF and GF. Empirically, we find that a model trained with vanilla cross-entropy loss is more robust to $G$ than a model trained with a CF-aware method. Motivated by this, we propose a simple baseline, named Counterfactual Knowledge Distillation (CKD), which distills the robustness to $G$ during the original CF-aware optimization. Finally, our extensive experiments using CelebA-CF and LFW-CF demonstrate that CF-achieving models satisfy GF when reducing the reliance on $G$ (*e.g.*, using CKD).

In summary, our contributions are three-fold. Firstly, we construct two new image classification benchmarks for measuring CF, CelebA-CF and LFW-CF. Secondly, using these datasets, we observe the disparity between CF and GF in image classifiers and provide a theoretical rationale; a counterfactually fair classifier may not necessarily achieve GF when an additional latent attribute that is correlated with the sensitive attribute exists. Finally, we propose a simple baseline, CKD, to reduce the sensitivity to such latent attributes of a model, resulting in achieving CF and GF simultaneously.

## 2    Constructing high-quality counterfactual images

The degree of counterfactual fairness (CF) can be measured by the prediction consistency between an original sample and its corresponding counterfactual (CTF) sample. For a given sample and a sensitive attribute, a CTF sample is defined as the one of which the sensitive attribute is altered while all the other attributes not caused by the sensitive attribute remain the same. However, acquiring a CTF sample for an image is challenging. For example, if the sex of a person in an image is the sensitive attribute, obtaining a CTF sample requires changing the secondary sex characteristics of the person such as beard or hairline, while preserving their identity and the other attributes, which is impossible in practice. One possible alternative is to generate a virtual face by altering such secondary sex characteristics of the given identity using a high-quality image editing method.

Several previous approaches [19, 34, 41, 44, 22] have attempted to generate CTF images by VAE or GAN-based editing methods. However, they have struggled with low image quality or unintended modifications to non-sensitive attributes, rendering them unreliable for evaluating CF. To address such issues, we employ IP2P [2], an advanced diffusion model-based image editing method. Notably,

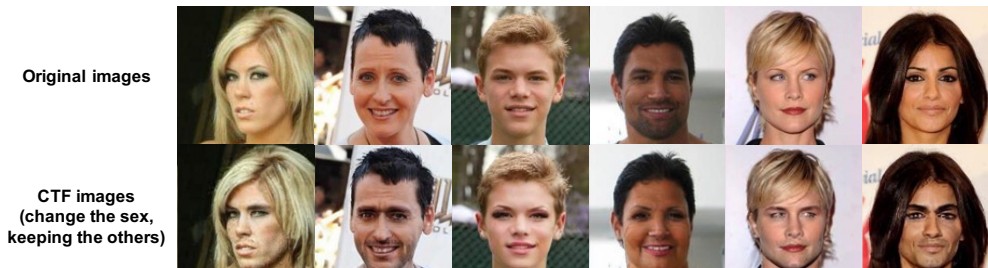

Figure 1: **CelebA-CF examples**. The counterfactual (CTF) images regarding the "sex attribute" are shown.

IP2P can generate high-quality CTF samples by simply adjusting the text instructions without any model retraining.

As the first step, we edit the test images of two popular facial image datasets, CelebA [25] and LFW [13]. We choose the "sex" of a person in an image as the sensitive attribute[3] and edit the sex-related visual characteristics of facial images using text prompts. We generated 720 CelebA CTF image pairs and 632 LFW CTF image pairs, where the images are selected to be balanced across groups for both target and sensitive labels. Here, we treat "blond hair" and "smiling" as the target labels for CelebA and LFW, respectively. Namely, for example, the CelebA CTF image pairs have a balanced group of <female, non-blond hair>, ..., and so on. Figure 1 and A.1 show examples of generated CTF images together with the originals. Hyperparameter settings are reported in Appendix C.1. Note that while we adopt the "sex" attribute, our generation process is attribute-agnostic (*e.g.*, age or skin color can be also used in place of sex) as illustrated in Figure A.2.

**Image filtering**. Despite the high quality of IP2P, low-quality CTF images can still be generated. To address this, we employed five human annotators to filter the images, *i.e.*, each image pair was annotated as either "correct" or "incorrect". To ensure objective and precise annotation criteria, we created guidelines as follows. Initially, we compiled a list of 20 masculine and feminine visual features using GPT-4o and with guidance from experts specialized in fairness, selected nine key facial attributes representing sex-related visual characteristics: facial hair, Adam's apple, skin texture, jawline, chin shape, brow ridge, cheekbone prominence, lip fullness, and hairline. These attributes were used to establish the criteria for evaluating correct CTF samples. One notable issue is that most of the feminine-like images in CelebA and LFW datasets include makeups (for instance, many female celebrities in the CelebA dataset appear to be wearing makeup) and the IP2P model is biased towards removing makeup when altering feminine features. To prevent images from being filtered out solely due to changes in makeup, we additionally included makeup in the set of key attributes, even though it is not a sex characteristic. Finally, the guidelines were created based on these ten attributes, providing some criteria for correct CTF samples, such as whether the change of some of the ten attributes was accurate and whether other facial characteristics remained consistent with the original image. Using these guidelines, we filtered out pairs receiving two or fewer "correct" votes, resulting in 230 and 144 images for CelebA and LFW, respectively. More details about the human annotating interface are in Appendix C.2, and additional information on the newly created dataset can be found in Appendix C.4.

**Reliability check**. We further verify the quality of our datasets by additional five human annotators, distinct from those participated in the filtering process. Those annotators evaluate only the images that remained after the filtering, based on two criteria: (1) whether the sensitive attribute was correctly changed and (2) whether the other non-sensitive attributes were preserved. The annotators evaluated the images for the sensitive attribute, "sex" and three non-sensitive attributes, "blond hair", "gray hair",

Table 1: **Human evaluation of the reliability of our datasets**. Accuracies of the correctly altered sensitive attributes and well-preserved non-sensitive attributes are shown.

|  | Sensitive | Non-Sensitive |
| --- | --- | --- |
| CelebA-CF | 96.52 | 95.98 |
| LFW-CF | 98.61 | 93.75 |

and "smiling"; we chose these three because other attributes can be subjective (*e.g.*, "big nose") [40] or had already been filtered (*e.g.*, "wearing hat"). Details of the annotating interface provided to

---

[3]The two datasets use the terms "gender" for indicating their sensitive attributes. However, using such terminology can present some ethical concerns because they can suggest meanings linked to social identities. Thus, we have decided to use the term "sex" instead, which more accurately refers to biological characteristics.

the five annotators are in Appendix C.2. Based on the majority vote, we compute the percentage of CTF samples which met each of the two criteria, *i.e.*, the accuracies for whether the sensitive and non-sensitive attributes are correctly altered and preserved. Table 1 displays the values for CelebA-CF and LFW-CF. The non-sensitive accuracy is averaged across three non-sensitive attributes. The results demonstrate that our CTF samples almost meet the two CTF criteria, suggesting that our datasets can be reliably utilized to evaluate CF.

**Ethical considerations**. In our study, we use the term "sex", not "gender", to represent the sensitive attribute with biological traits, because terms such as "gender" might imply associations with social identities, potentially raising some ethical issues. We also specifically choose ten perceived facial attributes as the visual features representing the biological sex in facial images. We believe that these considerations help alleviate various normative harms that arise from dichotomizing gender, which refers to social identity. However, despite our efforts, the sex-related visual characteristics are complex and intertwined, making it challenging to fully represent with a binary label. Thus, we urge practitioners to use our datasets with these considerations in mind.

## 3 Primitive study on the relationship between CF and GF

### 3.1 Experimental setup

We consider the image classification task where each data sample consists of an input image $X$, a class attribute $Y \in \mathcal{Y} = \{0, \cdots, |\mathcal{Y}| - 1\}$ and a sensitive attribute $A \in \{0, 1\}$, *e.g.*, sex.

**Metrics.** We measure three metrics for CF, GF, and classification accuracy. Firstly, we describe the metric for CF. A classifier satisfies CF when the predictions for the original sample and its counterfactual (CTF) sample are the same for every sample $x$ and sensitive attribute $a$, *i.e.*, $P(\widehat{Y} = y | X = x, A = a) = P(\widehat{Y}_{A \leftarrow a'} = y | X = x, A = a)$, where $\widehat{Y}_{A \leftarrow a'}$ represents the prediction for a counterfactual sample intervened on $A$ with $a'$ (*e.g.*, changing female to male). We quantify the degree of violence with respect to CF using counterfactual disparity (CD):

$$\text{Counterfactual Disparity (\textbf{CD})} \triangleq \mathbb{E}_{x,a}\big[P\big(\mathbb{1}\{\widehat{Y}_{A \leftarrow a'} \neq \widehat{Y}\} | x, a\big)\big]. \tag{1}$$

Secondly, we adopt equalized odds (EO) as our notion for GF. If a predictor $\widehat{Y}$ and the sensitive attribute $A$ are conditionally independent given the true class attribute $Y$, the predictor satisfies EO; namely, EO holds when $P(\widehat{Y} = y' | A = 0, Y = y) = P(\widehat{Y} = y' | A = 1, Y = y)$. From the definition, we can capture the degree of violence with respect to GF with the disparity of EO (DEO):

$$\text{Disparity of EO (\textbf{DEO})} \triangleq \max_{y, y' \in \mathcal{Y}} \big| P(\widehat{Y} = y' | A = 0, Y = y) - P(\widehat{Y} = y' | A = 1, Y = y) \big|. \tag{2}$$

We note that we empirically compute CD and DEO, defined in Equation (1) and (2), using our benchmark datasets and the original test datasets of CelebA and LFW, respectively. Additionally, Pinto et al. [33] propose several other metrics to evaluate CF, and accordingly, we conducted an additional evaluation based on these metrics, with results provided in Appendix G.5.

**Baseline methods.** We evaluate a model trained with the vanilla cross-entropy loss (denoted as "Scratch") and two CF-aware training methods, Scratch+aug and counterfactual pairing (CP). Scratch+aug is a Scratch method using an augmented training dataset with counterfactual samples [7], and CP [36] adopts a regularization term that promotes pairs of original and its CTF sample to obtain the same prediction (see Equation (4) for the formal definition). Note both methods need counterfactual samples for training, and hence, we use the samples generated via IP2P with the same prompts used in Section 2 without any filtering process to obtain results for them. For a comprehensive study, we additionally evaluate two individual fairness-aware methods, SenSeI [42] and LASSI [31], of which goals are analogous to CF in aiming to make a model robust to perturbation of the sensitive attribute. More details are described in Appendix D.3.

**Model selection.** Due to the accuracy-fairness trade-off [6], appropriate model selection is important for fair evaluation. We explore varying hyperparameters and select the best model that shows the lowest CD (Equation (1)) for the held-out validation set while achieving a lower bound of the accuracy[4].

---

[4]Considering the accuracy degradation of fair-training methods, we set the bound as 98% of the accuracy of Scratch, *i.e.*, if Scratch achieves 95.0% accuracy, then we only consider models with more than 93.1% accuracy.

Table 2: **CF does not always imply GF on image classification**. We report CD (Equation (1)) and DEO (Equation (2)) for measuring Counterfactual Fairness (CF) and Group Fairness (GF), respectively. Accuracy and DEO are measured on the original test datasets (CelebA and LFW) and CD is evaluated on the newly constructed datasets, CelebA-CF and LFW-CF, described in Section 2. If a model shows an inferior metric value than Scratch, the number is highlighted in red.

| Method | CelebA (and CelebA-CF) | | | LFW (and LFW-CF) | | |
|---|---|---|---|---|---|---|
| | Acc ↑ | CD ↓ | DEO ↓ | Acc ↑ | CD ↓ | DEO ↓ |
| Scratch | 95.53 | 10.26 | 47.10 | 90.85 | 18.06 | 7.66 |
| Scratch+aug [7] | 95.41 | 4.65 | 44.71 | 90.34 | 12.15 | **7.86** |
| CP [36] | 94.10 | 2.53 | **51.01** | 89.77 | 9.20 | **8.74** |
| SenSeI [42] | 95.33 | 8.00 | **52.32** | 87.75 | 16.09 | **9.23** |
| LASSI [31] | 91.07 | 9.69 | 31.79 | - | - | - |

## 3.2 Performance comparison

Table 2 shows accuracy, CD, and DEO for Scratch and four baseline methods. Note that we omit the result of LASSI on LFW because the number of samples in LFW is not enough to train the Glow model [20], which is the main component of LASSI. From the table, CF-aware and individual fairness-aware methods are mostly effective in mitigating CD, when compared to Scratch. However, it does not necessarily lead to improvements in DEO. Especially, while CP significantly improves CD for both datasets, it exacerbates DEO compared to Scratch. Namely, contrary to the previous studies [1, 35] showing that CF implies GF on tabular datasets, our observation shows that CF does not always imply GF on image datasets. In the following section, we theoretically investigate why the previous observations may not hold on images.

# 4 Theoritical analysis on the relationship between CF and EO for images

## 4.1 Structural Causal Model (SCM) for images

Structural Causal Models (SCMs) are represented as directed acyclic graphs satisfying the conditions specified in [30]. In these models, nodes and edges indicate variables and their causal relationships within the data-generating process. As studied in previous works [4, 22], the nodes of an SCM for image can be categorized into three parts. As shown in Figure 2, the blue, gray and yellow nodes indicate latent attributes, *e.g.*, $Y$ or $A$, components of the image influenced by these attributes, *e.g.*, $X_Y$ or $X_A$, and the whole image $X$, respectively. Taking an SCM for facial images as an example, we can interpret these nodes as follows: latent attributes such as hair color or sex, facial components like the hair or an Adam's apple in a facial image, and an entire face. Note that the blue region in the figure describes that various causal relationships among latent attributes can exist [5]. Furthermore, although an image SCM may contain additional latent attributes, we simplify our focus to only include the class and sensitive attribute, $Y$ and $A$, and a third-party attribute, $G$, which may correlate with the sensitive attribute $A$.

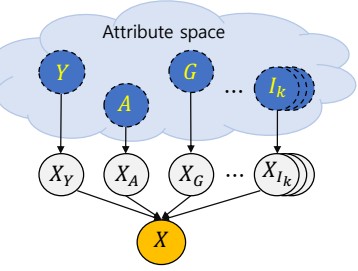

Figure 2: **Image SCM**. Blue, gray, and yellow circles represent latent attributes, components of an image and a whole image, respectively. Directed edges indicate a causal relationship from the source to the target. The blue region indicates that there can be any direction of edges between blue nodes.

## 4.2 Theoretical analysis

According to the Markov assumption of SCM [30], if there are no unblocked paths between two variables in an SCM (*i.e.*, they are d-separated), the variables are statistically independent. Utilizing this property, Anthis and Veitch [1] demonstrated that CF implies several GF notions, including Equalized Odds (EO) [10], under the specific condition on SCMs such as no *backdoor* path from the

---

[5]We assume no edge or unblocked path from $A$ to $Y$; otherwise, all counterfactually fair models based on that SCM would produce random predictions with respect to $X_Y$.

sensitive attribute $A$ to the image $X$ exists (Theorem 2 of [1]). Moreover, the authors empirically show that these conditions would hold on some tabular datasets.

However, we argue that these conditions would not hold for image datasets due to a fundamental difference in what sensitive attributes represent in an image. Specifically, tabular datasets typically consist of recorded information by subjects, where sensitive attributes such as sex or race usually represent immutable genetic information; hence they are not caused by other attributes and cause all attributes correlated with the sensitive attributes. In contrast, sensitive attributes in image datasets indicate visual characteristics that can change and be influenced by some other attributes, such as the attribute $G$. For example, in a facial image dataset, attributes like hair length or accessories might be highly correlated with, but not caused by, secondary sex characteristics such as beard. Namely, a backdoor path from the sensitive attribute $X$ through the attribute $G$ could exist, thereby breaking the connection between CF and GF discovered in previous studies.

Our theoretical result specifies the relationship between CF and GF (especially for EO) with $G$:

**Theorem 4.1.** *Assume a latent attribute $G$ in Figure 2 is a non-descendant variable of $A$ and connected to $A$ through an unblocked path. Then, the following inequality holds for a counterfactually fair classifier $\boldsymbol{\theta}$ and any pairs of $y$ and $y'$:*

$$\left| P(\widehat{Y} = y'|A = 0, Y = y) - P(\widehat{Y} = y'|A = 1, Y = y) \right|$$
$$\leq \sum_{X_Y} P(X_Y|Y = y) \max_{X_G, X_G'} d_{\boldsymbol{\theta}, X_Y}(X_G, X_G'), \qquad (3)$$

*in which $d_{\boldsymbol{\theta}, X_Y}(X_G, X_G') = \left| P(\widehat{Y} = y'|X_Y, X_G) - P(\widehat{Y} = y'|X_Y, X_G') \right|$ and $\hat{Y}$ is the prediction of the model $\boldsymbol{\theta}$. The equality holds when $d_{\boldsymbol{\theta}, X_Y} = 0$ always regardless of $X_Y$.*

The proof of the theorem is in Appendix A. Note that when we take the maximum over $(y, y')$ on both sides of the inequality in Theorem 4.1, the left-hand side of the inequality becomes identical to DEO (Equation (2)). Therefore, the theorem implies that DEO is upper bounded by the maximum of $d_{\boldsymbol{\theta}, X_Y}(X_G, X_G')$ (in which the maximum is over $X_G, X_G', y, y'$), which measures the sensitivity of the model with respect to $G$. In other words, the theorem shows that when a counterfactually fair model is sensitive to $X_G$ (*i.e.*, when $\max d_{\boldsymbol{\theta}, X_Y}(X_G, X_G')$ is large), the model may result in having high DEO in the worst-case.

Theorem 4.1 elucidates why CF-aware methods in Table 2 often fail to mitigate DEO despite significant improvements in CD. Namely, if the attribute $G$ assumed in Theorem 4.1 exists on CelebA and LFW, DEO for the classifiers trained by CF-aware methods can worsen depending on their robustness to $G$. This will be empirically demonstrated using "hair length" as $G$ in Section 5.2, together with the results using a controllable synthetic dataset. Furthermore, Theorem 4.1 suggests that we can re-establish the relationship between two notions by making counterfactually fair classifiers non-sensitive to $G$. In the following section, we introduce a method to promote a classifier not to depend on $G$ while achieving CF.

## 5 Empirical analyses on the effect of $G$ to CF and GF

### 5.1 Counterfactual Knowledge Distillation (CKD)

Motivated by Theorem 4.1, we propose a baseline fair-training method to achieve both CF and GF. Conceptually, if we can reduce the dependency between the latent attribute $G$ described in Theorem 4.1 and the prediction of a CF-aware trained model, we can expect that the model will achieve CF and GF simultaneously. Therefore, we improve the CF-aware method, CP [36] (best-performing in Table 2), such that the dependency to the attribute $G$ is reduced. We first describe the CP regularization (which is used along with the cross-entropy loss) for given counterfactual samples $\mathcal{D}' = \{x_{i, A \leftarrow a_i'}\}_{i=1}^N$ corresponding to the original training dataset $\mathcal{D}$:

$$\mathcal{L}_{\text{CP}}(\theta, \mathcal{D} \cup \mathcal{D}') := \frac{1}{N} \sum_{i=1}^N \|f(\boldsymbol{\theta}, x_i) - f(\boldsymbol{\theta}, x_{i, A \leftarrow a_i'})\|_2^2, \qquad (4)$$

in which $f(\boldsymbol{\theta}, x)$ is a representation vector of input $x$ produced by a classifier $\boldsymbol{\theta}$, such as logit or feature vector. Note that the images $x$ and $x_{A \leftarrow a'}$ differ only in their components corresponding to

the sensitive attribute $A$ and the attributes caused by the sensitive attribute $A$. Hence, although the CP regularization works well for achieving CF, it does not ensure the model does not rely on the attribute $G$, potentially leading to worse DEO as argued in the previous section.

Recent studies [16, 38, 45] have shown that the robustness of a teacher model can be transferred into a student model through knowledge distillation (KD) [12]. To that end, we first assume a teacher model that is robust to the attribute $G$ is available. Then, our idea is to apply both KD and CP regularization to train our student model, which leads to a simple yet effective approach, dubbed as Counterfactual Knowledge Distillation (CKD). Specifically, CKD employs averaged representation vectors of original and counterfactual samples extracted by the teacher model $\boldsymbol{\theta}^T$ as target vectors. Then, representation vectors of both samples from the student model $\boldsymbol{\theta}$ are enforced to follow the target vectors. Namely, the distillation term of CKD is defined as follows:

$$\mathcal{L}_{CKD}(\boldsymbol{\theta}, \mathcal{D} \cup \mathcal{D}') := \frac{1}{2N} \sum_i^N \left( \|f(\boldsymbol{\theta}, x_i) - f_i^T\|_2^2 + \|f(\boldsymbol{\theta}, x_{i, A \leftarrow a_i'}) - f_i^T\|_2^2 \right),$$

in which $f_i^T = \frac{1}{2}\big(f(\boldsymbol{\theta}^T, x_i) + f(\boldsymbol{\theta}^T, x_{i, A \leftarrow a_i'})\big)$ is the target vector for the $i$-th pair. (5)

Note that our distillation terms have both effects of KD and CP by promoting both representations of original and counterfactual samples to be aligned with the target vectors $f_i^T$ produced by the teacher model. Therefore, based on Theorem 4.1, we can deduce that the CKD regularization encourages the model to achieve both CF (by the CP effect) and EO-based GF (by the KD effect that distills the robustness of the teacher with respect to the attribute $G$). In addition, we optionally incorporate CP regularization into our objective to further mitigate CD. The final objective of our method (which we again dub as CKD for brevity) is as follows:

$$\min_{\boldsymbol{\theta}} \mathcal{L}_{\text{CE}}(\boldsymbol{\theta}, \mathcal{D}) + \mu \mathcal{L}_{\text{CKD}}(\boldsymbol{\theta}, \mathcal{D} \cup \mathcal{D}') + \lambda \mathcal{L}_{\text{CP}}(\boldsymbol{\theta}, \mathcal{D} \cup \mathcal{D}'), \quad (6)$$

in which $\mu$ and $\lambda$ are controllable hyperparameters for the CKD and CP regularization, respectively.

While we assumed above the availability of a teacher model that is robust to the attribute $G$, obtaining such a model could be challenging in practice. Empirically, we observe that vanilla-trained models (referred to as "Scratch" models) less depend on the attribute $G$ than CP-trained ones (see Figure 3 and Table 3 for more details). We presume that this is because the attributes $A$ and $G$ behave as "shortcut" features for classifying the class attribute $Y$, *i.e.*, they are easy-to-learn discriminatory features. As observed by Scimeca et al. [37], making a model blind to a certain shortcut feature causes it to rely more heavily on the other shortcut features. In our case, CP-trained models are trained to be invariant to the sensitive attribute $A$, resulting in a greater dependence on the attribute $G$ compared to the Scratch models. Thus, unless otherwise specified, we will assume the vanilla-trained model is relatively robust to the attribute $G$ since it would mostly rely on the sensitive attribute $A$, hence, we use it as the teacher model.

## 5.2 Impact of robustness to $G$ on CF and GF

We empirically validate our theoretical result and CKD on both a newly introduced synthetic dataset (CIFAR-10B) and a real-world dataset (CelebA) by analyzing CF, GF, and the robustness with respect to the attribute $G$ described in Theorem 4.1. We thus introduce a new metric for the robustness to the attribute $G$, the rate of flipped predictions (**RFP**) :

$$\text{RFP} \triangleq \mathbb{E}_{x,x'}\big[P\big(\mathbb{1}\{\widehat{Y} \neq \widehat{Y}'\}|x, x'\big)\big]. \quad (7)$$

in which $x$ is an original image, $x'$ is its corresponding image with the attribute $G$ flipped. $\widehat{Y}$ and $\widehat{Y}'$ refer to the predicted label by the trained model given $x$ and $x'$, respectively. RFP quantifies the amount of flipped predictions when the attribute $G$ is altered. For example, if a model shows the same prediction after changing the attribute $G$, its RFP becomes 0%.

**CIFAR-10B, a controllable synthetic dataset.** We construct the CIFAR-10B dataset, where we can perfectly control the degree of bias with respect to the attribute $G$ while the target label is biased towards the sensitive attribute $A$. We make binary class labels from the 10 classes of CIFAR-10 (0-4 and 5-9 classes). We set the attributes $A$ and $G$ in Theorem 4.1 with the presence of Gaussian and Contrast noise, respectively. We also set a fixed ratio of 0.8 and a controllable ratio $\alpha$, which represent

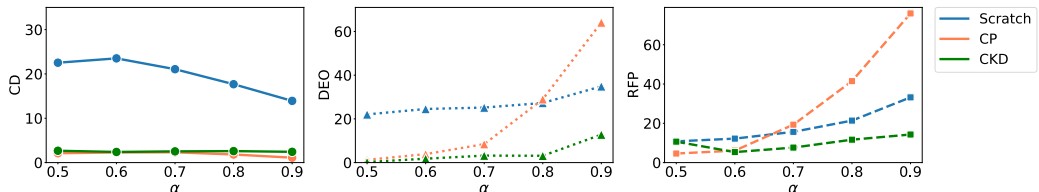

Figure 3: **Impact of the correlation of $G$ and $A$.** $\alpha$ indicates how $A$ and $G$ are correlated on CIFAR-10B.

skewnesses among $(Y, A)$ and $(A, G)$, respectively; the former ratio is the spurious correlation between $Y$ and $A$, and the latter one is the correlation between $A$ and $G$. We then construct the CIFAR-10B dataset by randomly injecting Gaussian or Contrast noise to each CIFAR-10 image at given ratios, as illustrated in Figure E.2. Unless otherwise noted, we set $\alpha$ as 0.8.

We train models with Scratch, CP, and CKD on CIFAR-10B by adjusting $\alpha$ from 0.5 (*i.e.*, $A$ and $G$ are decorrelated) to 0.9 at intervals of 0.1. Figure 3 shows CD, DEO, and RFP metric values for each method. The figures indicate that while CP and our CKD consistently achieve CF, CP fails to meet GF as $\alpha$ increases, potentially due to higher RFP. Furthermore, RFP of Scratch is lower than that of CP when $\alpha$ is greater than 0.7. This empirically justifies the use of vanilla-trained models as teacher models robust to $G$. By using these teacher models, CKD significantly improves DEO regardless of the value of $\alpha$ by maintaining the robustness to $G$, *i.e.*, low RFP, supporting the result of Theorem 4.1.

**Impact of $G$ manipulation on CelebA.** We assume "hair length" as $G$ for facial image datasets, *e.g.*, CelebA because the hair length $G$ can be highly correlated with, but not caused by, the sex $A$. To compute RFP for the hair length attribute, we manipulate the hair length of CelebA test images using SDEdit [28]. More details and generated examples can be found in Appendix E.1. Using the hair length-edited images, we report RFP in Table 3, together with DEO and CD. The results share the same trend as the CIFAR-10B results, *i.e.*, CP shows worse DEO and RFP than Scratch but better CD, whereas CKD shows the best DEO and RFP, despite a slight increase of CD.

Table 3: **Impact of $G$ on CelebA.** We assume "hair length" as $G$ and manipulate the hair length of test images. CD, DEO, and RFP are measured on CelebA-CF, CelebA, and hair-edited CelebA, respectively.

| Method | | CelebA | |
| | CD ↓ | DEO ↓ | RFP ↓ |
| --- | --- | --- | --- |
| Scratch | 10.26 | 47.10 | 15.27 |
| CP | **2.53** | 51.01 | 20.37 |
| CKD | 4.44 | **13.23** | **10.85** |

## 5.3 Impact of the robustness to $G$ of the teacher model on CKD

Our CKD requires a robust teacher model with respect to the attribute $G$ to distill the robustness to the target model. To analyze the impact of the robustness of the teacher model, we compare various teacher models with different dependencies on the attribute $G$ using CIFAR-10B. We consider four teacher models, ordered by robustness to the attribute $G$: CP ($\boldsymbol{\theta}_{\text{CP}}^T$), Scratch ($\boldsymbol{\theta}_{\text{Scratch}}^T$), and CKD model with a Scratch teacher ($\boldsymbol{\theta}_{\text{CKD}}^T$), and a de-biased model trained on CIFAR-10B balanced for $G$, *i.e.*, $\alpha = 0.5$, ($\boldsymbol{\theta}_{\text{De-biased}}^T$). Using these teacher models, we report DEO, CD, and RFP of CKDs on the CIFAR-10B dataset in Table 4. We observe that the degree of robustness to the attribute $G$ of the teacher model (*i.e.*, $\text{RFP}^T$) highly correlates to DEO. It is because as the teacher model becomes more robust to $G$, RFP of the target model

Table 4: **Impact of robustness to $G$ of the teacher model.** $\boldsymbol{\theta}_{\text{CP}}^T$, $\boldsymbol{\theta}_{\text{CKD}}^T$, and $\boldsymbol{\theta}_{\text{Scratch}}^T$ are CP, CKD, and Scratch teacher model. $\boldsymbol{\theta}_{\text{De-biased}}^T$ is a Scratch model trained on a perfectly de-biased training dataset ($\alpha = 0.5$). $\text{RFP}^T$ denotes how a teacher is biased towards $G$. CD, DEO, RFP are metrics for evaluating CF, GF, and bias towards $G$, respectively. Results are measured on CIFAR-10B with $\alpha = 0.8$

| Method | $\text{RFP}^T$ ↓ | Acc ↑ | CD ↓ | DEO ↓ | RFP ↓ |
| --- | --- | --- | --- | --- | --- |
| CKD w/ $\boldsymbol{\theta}_{\text{CP}}^T$ | 41.46 | 76.15 | 3.59 | 12.65 | 18.08 |
| CKD w/ $\boldsymbol{\theta}_{\text{Scratch}}^T$ | 21.38 | 78.49 | 2.85 | 7.30 | 11.66 |
| CKD w/ $\boldsymbol{\theta}_{\text{CKD}}^T$ | **11.66** | 78.39 | **2.33** | **4.89** | **5.30** |
| CKD w/ $\boldsymbol{\theta}_{\text{De-biased}}^T$ | **10.81** | 77.17 | **2.79** | **4.01** | **4.67** |

gets lower, finally leading to a lower DEO while maintaining fair CD. Namely, these results support our theoretical result again.

Table 5: **Evaluation of GF and CF of fair-training for image classification**. The details are the same as Table 2. "Scratch" denotes a model trained without considering the notion of fairness through the vanilla cross-entropy loss. "+aug" denotes counterfactual (CTF) image augmentation described in Section 3. If a model performs worse than the Scrath model on CD/DEO, we highlight the numbers in red. The best performance is highlighted in orange, and the second-best performance is highlighted in grey.

| Method | CIFAR-10B ($\alpha$=0.8) | | | CelebA (and CelebA-CF) | | | LFW (and LFW-CF) | | |
|---|---|---|---|---|---|---|---|---|---|
| | Acc ↑ | CD ↓ | DEO ↓ | Acc ↑ | CD ↓ | DEO ↓ | Acc ↑ | CD ↓ | DEO ↓ |
| Scratch | 78.01 | 17.90 | 27.46 | 95.53 | 10.26 | 47.10 | 90.85 | 18.06 | 7.66 |
| SS [14] | 74.77 | 16.42 | 25.73 | 95.44 | 9.13 | 42.95 | 90.43 | *18.19* | 6.75 |
| RW [17] | 76.53 | 12.15 | 18.94 | 95.16 | 5.50 | 24.21 | 90.87 | *18.68* | 6.92 |
| COV [43] | 79.03 | 13.90 | 24.05 | 94.42 | 7.72 | 34.04 | 90.85 | 16.43 | 6.99 |
| MFD [16] | 76.84 | 12.24 | 15.39 | 94.37 | 4.61 | 19.00 | 90.47 | 16.07 | 2.15 |
| LBC [15] | 76.16 | 15.01 | 17.12 | 94.92 | 6.24 | 22.61 | 90.71 | 15.76 | 3.56 |
| SS+aug | 73.45 | 9.95 | 15.21 | 95.17 | 5.24 | 40.80 | 89.96 | 15.23 | 6.82 |
| RW+aug | 76.15 | 12.93 | 20.94 | 95.13 | 5.34 | 24.63 | 90.76 | *18.63* | 6.71 |
| COV+aug | 76.52 | 8.17 | 15.04 | 94.08 | 8.11 | 29.03 | 90.47 | 13.65 | 6.78 |
| MFD+aug | 77.10 | 11.16 | 14.79 | 93.78 | 3.87 | 14.36 | 89.90 | *19.36* | 2.47 |
| LBC+aug | 75.82 | 9.01 | 15.29 | 94.39 | 9.32 | 36.08 | 88.66 | 12.41 | 2.79 |
| CP [36] | 75.26 | **2.05** | *33.23* | 94.10 | **2.53** | *51.01* | 89.77 | 9.20 | *8.74* |
| SS+CP | 76.54 | 3.14 | **9.08** | 94.54 | **2.40** | 37.97 | 88.7 | **6.13** | 4.26 |
| RW+CP | 75.68 | 8.83 | 13.92 | 95.19 | 4.67 | 25.56 | 90.87 | 15.24 | 6.16 |
| COV+CP | 77.74 | 4.30 | 19.42 | 94.29 | 5.36 | *51.63* | 91.23 | 11.91 | 6.52 |
| MFD +CP | 76.67 | 10.01 | 13.17 | 93.81 | 3.47 | 23.31 | 89.39 | 15.15 | **1.90** |
| LBC+CP | 76.88 | 3.02 | 12.45 | 95.12 | 4.72 | 22.78 | 89.92 | 8.33 | 3.02 |
| CKD ($\lambda = 0$) | 76.32 | 8.59 | 11.23 | 94.12 | 4.31 | **14.11** | 90.76 | 12.42 | 2.64 |
| CKD | 78.49 | **2.85** | **7.30** | 93.08 | 4.44 | **13.23** | 89.26 | **7.94** | **1.88** |

# 6 Full comparisons of fair-training methods on image classification

Finally, we evaluate the existing fair-training methods focusing on group fairness (GF) and counterfactual fairness (CF) on CelebA and LFW, together with CIFAR-10B for image classification tasks. We emphasize that only CelebA-CF and LFW-CF have counterfactual images of the real-world images; hence, we measure a CF metric, *i.e.*, Counterfactual Disparity (CD) (Equation (1)), using our datasets. Along with the CF-aware methods, such as CP [36] and CKD, we report the GF-aware methods including SS [14], RW [17], COV [43], MFD [16], and LBC [15]. In addition, we report the naive combinations of GF-aware and CF-aware methods, *e.g.*, training GF-aware method with the augmented training dataset with counterfactual images generated by IP2P [2] (denoted as "+aug") and combinations of the GF-aware methods and the CP regularization (Equation (4)) (denoted as "+CP"). The hyperparameters for all methods besides the GF-aware methods are selected using the same protocol in Section 3, and ones for the GF-aware methods are chosen based on DEO using the same lower bound of the accuracy. Implementation details are provided in Appendix D.3.

Table 5 shows the holistic evaluation of CF and GF for all the methods mentioned above on the three image classification tasks. The table shows four important observations. First, although CP (a CF-aware method) mostly performs the best on CD, it even shows worse DEO than Scratch. We theoretically and empirically discussed the reason in Section 4 and 5. Second, the GF-aware methods are effective in improving DEO but have a minimal impact on CD. This suggests that the faithfulness assumption for SCM may not hold, which will be discussed in more details in Appendix F. Third, the naive combinations of GF-aware and CF-aware methods exhibit much better CD than using the GF-aware methods alone. Additionally, DEOs achieved by the naively combined methods tend to be improved since their training datasets are balanced over the sensitive attributes by incorporating generated samples into the original training datasets. Lastly, we found that CKD shows the best DEO for every evaluation dataset. It shows that if we can train a CF-aware model by reducing the dependency on $G$, we can achieve both CF and GF even on the image classification task. We additionally conduct an ablation study on CKD by removing CP (*i.e.*, CKD ($\lambda = 0$)) from Equation (6). We observe that this ablated version achieves suboptimal performances than CKD.

This suggests that adding the CP regularization term to the CKD objective function can be helpful to improve both CD and DEO.

# 7 Concluding remarks

This paper offers carefully crafted benchmark datasets for evaluating the counterfactual fairness (CF) of image classification methods. Since obtaining true counterfactual images is impossible in practice, we employ a high-quality image editing technique to generate counterfactual images of the given images. We construct two facial image benchmarks, CelebA-CF and LFW-CF, by carefully filtering out and verifying the generated counterfactual images by human annotators. Our datasets relax the constraints of the impossibility of evaluating CF in image classification. Using our datasets, we also provide theoretical and empirical results showing that CF may not imply GF, contradictory to the studies conducted on tabular datasets. We elucidate this phenomenon by the presence of the third-party attribute highly correlated with, but not caused by, the sensitive attribute. From this finding, we propose a simple baseline method, CKD, to achieve CF and GF simultaneously. Our extensive experimental results on both GF and CF metrics show that when reducing the reliance on the attribute (*e.g.*, by using CKD), improving the CF metric leads to a significant improvement in the GF metric. By providing our benchmarks and various analyses, we believe that our findings bridge CF and GF in image classification, contributing to the development of fair and robust image recognition systems.

## Acknowledgments

This work was supported in part by the National Research Foundation of Korea (NRF) grant [No.2021R1A2C2007884] and by Institute of Information & communications Technology Planning & Evaluation (IITP) grants [RS-2021-II211343, RS-2021-II212068, RS-2022-II220113, RS-2022-II220959] funded by the Korean government (MSIT). It was also supported by AOARD Grant No. FA2386-23-1-4079, SNU-Naver Hyperscale AI Center, and Hyundai Motor Chung Mong-Koo Foundation.

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
