# A Proof of Theorem 4.1

We start from LHS in equation 3:

$$\left| P(\widehat{Y} = y'|A = 0, Y = y) - P(\widehat{Y} = y'|A = 1, Y = y) \right|$$

$$= \left| \sum_{X_A, X_Y, X_G} P(\widehat{Y} = y'|X_A, X_Y, X_G, A = 0, Y = y)P(X_A, X_Y, X_G|A = 0, Y = y) \right.$$

$$\left. - P(\widehat{Y} = y'|X_A, X_Y, X_G, A = 1, Y = y)P(X_A, X_Y, X_G|A = 1, Y = y) \right| \quad \text{(A.1)}$$

$$= \left| \sum_{X_A, X_Y, X_G} P(\widehat{Y} = y'|X_Y, X_G)P(X_A, X_Y, X_G|A = 0, Y = y) \right.$$

$$\left. - P(\widehat{Y} = y'|X_Y, X_G)P(X_A, X_Y, X_G|A = 1, Y = y) \right| \quad \text{(A.2)}$$

$$= \left| \sum_{X_Y, X_G} P(\widehat{Y} = y'|X_Y, X_G)\big(P(X_Y|X_G, A = 0, Y = y)P(X_G|A = 0, Y = y) \right.$$

$$\left. - P(X_Y|X_G, A = 1, Y = y)P(X_G|A = 1, Y = y)\big) \right| \quad \text{(A.3)}$$

$$= \left| \sum_{X_Y, X_G} P(\widehat{Y} = y'|X_Y, X_G)\Big( P(X_Y|Y = y)P(X_G|A = 0, Y = y) - P(X_Y|Y = y)P(X_G|A = 1, Y = y)\Big) \right|$$

$$\text{(A.4)}$$

$$= \left| \sum_{X_Y} P(X_Y|Y = y)\Big( \sum_{X_G} P(\widehat{Y} = y'|X_Y, X_G)P(X_G|A = 0, Y = y) \right. \tag{A.5}$$

$$\left. - \sum_{X_G'} P(\widehat{Y} = y'|X_Y, X_G')P(X_G'|A = 1, Y = y)\Big) \right|. \tag{A.6}$$

Note the first and third equalities are driven by Bayes' theorem, the second one is from the independence between $\widehat{Y}$ and $X_G, A$ conditioned on $X_Y, X_G$ based on the Markov properties of SCM, and the fourth one is due to the independence between $X_Y$ and $X_G, A$ conditioned on $Y$. We denote a coupling between the two distributions $P(X_G|A = 0, Y)$ and $P(X_G'|A = 1, Y)$ as $\Pi(X_G, X_G')$, then we have:

$$\left| P(\widehat{Y} = y'|A = 0, Y = y) - P(\widehat{Y} = y'|A = 1, Y = y) \right|$$

$$= \left| \sum_{X_Y} P(X_Y|Y = y)\Big( \sum_{X_G, X_G'} \Pi(X_G, X_G')\big(P(\widehat{Y} = y'|X_Y, X_G) - P(\widehat{Y} = y'|X_Y, X_G')\big)\Big) \right|. \tag{A.7}$$

$$\leq \sum_{X_Y} P(X_Y|Y = y)\Big( \sum_{X_G, X_G'} \Pi(X_G, X_G')\big| P(\widehat{Y} = y'|X_Y, X_G) - P(\widehat{Y} = y'|X_Y, X_G')\big| \Big) \tag{A.8}$$

$$= \sum_{X_Y} P(X_Y|Y = y) \sum_{X_G, X_G'} \Pi(X_G, X_G')d_{\boldsymbol{\theta}, X_Y}(X_G, X_G') \tag{A.9}$$

where the sample distance is denoted as $d_{\boldsymbol{\theta}, X_Y}(X_G, X_G') = \left| P(\widehat{Y} = y'|X_Y, X_G) - P(\widehat{Y} = y'|X_Y, X_G') \right|$. The inequality in Equation A.8 is driven by Jensen's inequality.

# B Limitations and societal impacts

While our datasets and analyses reveal the relationship between CF and GF in image classification, we clarify our study's limitations. First of all, our study uses sex as a sensitive attribute based on visually perceived biological traits. However, as mentioned in Section 2, this simplification does not capture the full spectrum of sexual traits, which is more complex and nuanced. Therefore, we emphasize again that practitioners should use our data with these considerations in mind; they should not utilize our datasets for gender categorization but rather for investigating the unfairness in terms of CF and GF and enhancing fairness in AI systems. Second, our data generation process relies on IP2P to create CTF samples. We tried to mitigate the potential bias problem during the data generation process through the sophisticated human filtering process, but our data samples could be

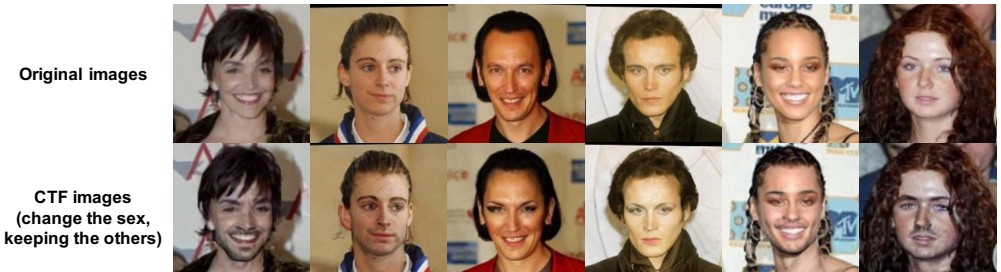

Figure A.1: **LFW-CF examples**. The counterfactual (CTF) images regarding the "sex attribute" are shown. The top row shows the original image, while the bottom row displays the CTF image generated by IP2P.

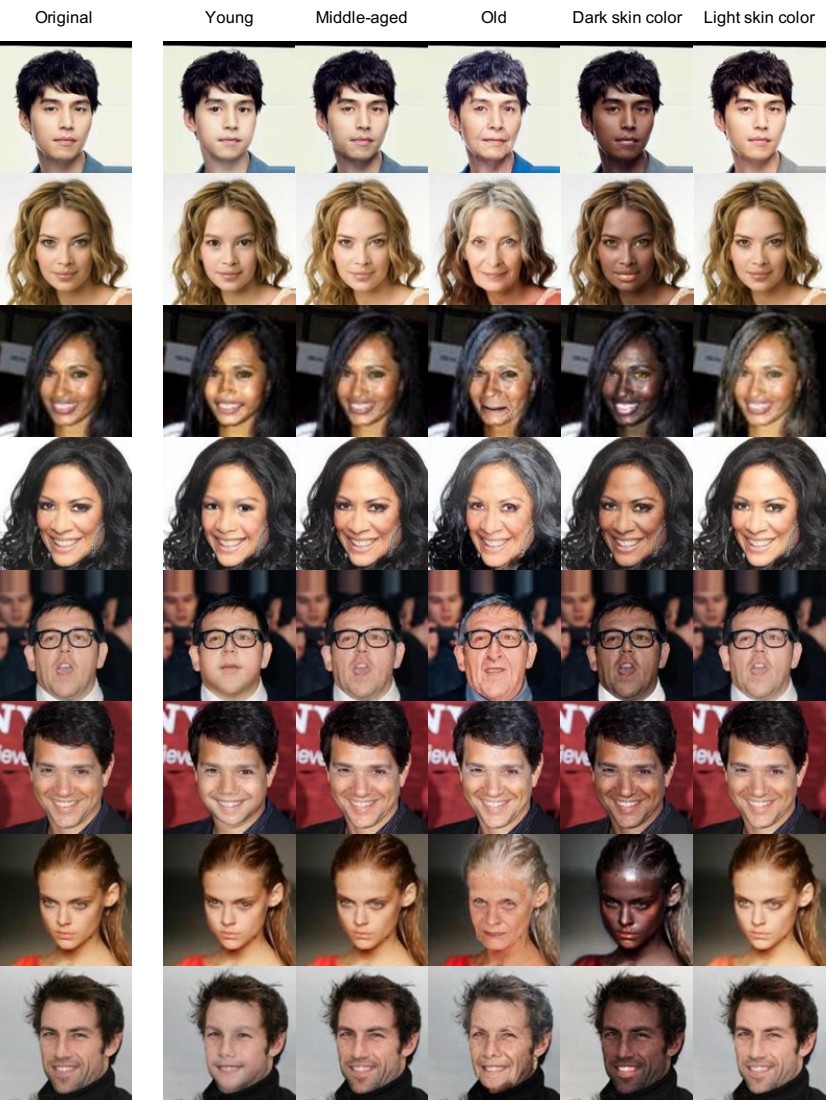

Figure A.2: **CF examples with other sensitive attributes**. Original and CTF samples are shown when age or skin color is considered as the sensitive attribute.

affected by the unintended bias of IP2P. Third, while we assume the structural causal model (SCM) for images as Figure 2, specifying an SCM in the real world is often infeasible. This difficulty also makes it challenging to apply some of our experiments, such as analyzing the attribute $G$ or using a robust teacher model to $G$. However,

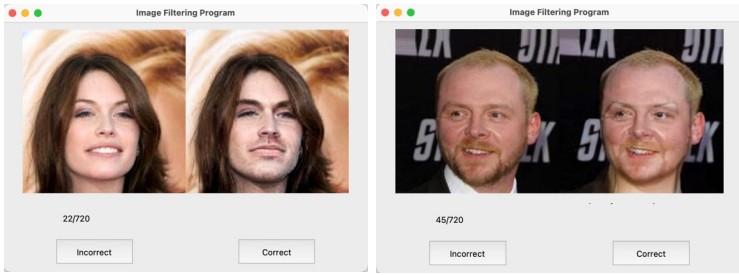

Figure C.1: User Interface shown to five annotators for image filtering.

we addressed these challenges to some extent by proposing a systematic method to investigate $G$ (Appendix E.2) and analyzing the robustness of vanilla teacher models (Section 5.3).

Despite the limitations, we believe that our work makes significant contributions through our curated image datasets and extensive analyses to the evolving field that addresses relationships among different fairness notions.

# C    Datasets: CelebA-CF and LFW-CF

We provide access to our newly created dataset, *i.e.*, CelebA-CF and LFW-CF, through the following link: CelebA-CF (https://figshare.com/s/62b6f7f69d0eab9c3c71) , LFW-CF (https://figshare.com/s/39f2daac58148e10e5fe)

## C.1    Hyperparameters for IP2P image editing

We set the resolution of generated images to 256×256 and the denoising step to 50. Furthermore, we set the Image-CFG weight to 1.8 and the Text-CFG weight to 7.5. These two hyperparameters are the guidance scales to control how the generated images closely resemble the input image or are intensely edited. To alter the sensitive attribute of facial images, we use the prompts of "turn the woman into a man" for female images and "turn the man into a woman" for male images.

## C.2    Human annotations

After generating CTF images using IP2P, we filtered them through five annotators to construct high-quality CTF samples, namely CelebA-CF and LFW-CF. Before evaluating the created CTF samples, the guidelines are given to the annotators, as presented in Figure C.2. To establish the guidelines, we extracted 20 facial attributes of secondary sex characteristics using Chat-GPT and then, with guidance from experts specialized in fairness, selected 9 key facial attributes (facial hair, Adam's apple, skin texture, jawline, chin shape, brow ridge, cheekbone prominence, lip fullness, and hairline). The guidelines instruct human annotators to filter out counterfactual samples based on these attributes (including considerations for the presence of makeup). Subsequently, given the original image and generated CTF image pair, annotators assess whether the generated image is correctly created based on the instructions. Figure C.1 shows the interfaces of the annotation task for image filtering.

We further verify the reliability of our datasets with another five human annotators, different from those who participated in the previous filtering process, and report the result in Table 1. For more objective annotation, we show 8 example images to the annotators before the labeling task, which are randomly sampled the same number of times for each attribute value from test image datasets. Then, the annotators label CelebA-CF and LFW-CF for 4 attributes, *i.e.*, "sex", "blond hair", "gray hair", and "smiling" in order. Specifically, the annotators evaluate whether the sensitive attribute was correctly altered and the non-sensitive attributes were maintained for a generated image. Similar to the image filtering task, we provide the annotators with the set of 10 sex-related facial attributes for objective and accurate labeling. Figure C.3 shows the interfaces of the annotation task for this reliability check. We provide the attribute values originally annotated for the original image datasets on the screen together for annotators to refer to as a guide for their annotating tasks. Although we focus on visually perceived sex traits, we use the terms Male and Female for convenience in the annotation interface.

We note that the wage paid to each participant is 18 USD per hour, resulting in a total expenditure of 360 USD on participant compensation.

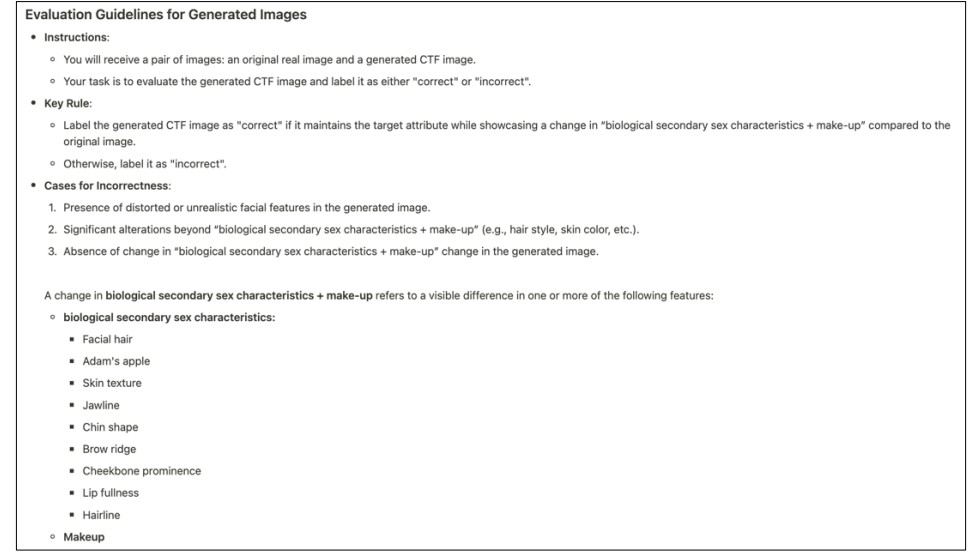

Figure C.2: The guideline instructions that were given to the five annotators for the image filtering.

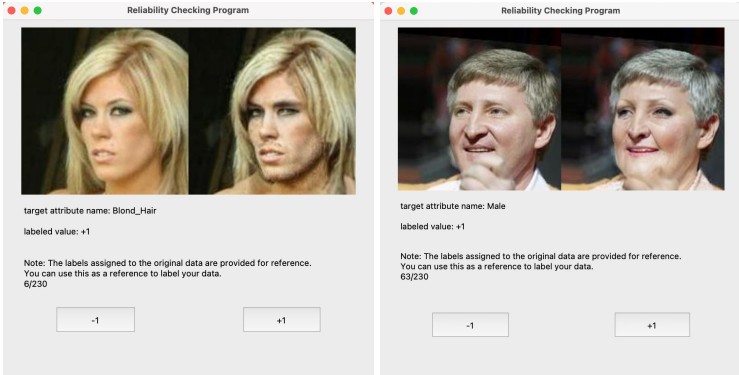

Figure C.3: User Interface shown to five annotators to evaluate the reliability of our created datasets.

## C.3 License information of assets employed in this study

- CelebA [25] was made available for academic research purposes without a formal license. The dataset can be downloaded at https://mmlab.ie.cuhk.edu.hk/projects/CelebA.html.

- LFW [13] is publicly available for research purposes. There is also no formal license and further information is reported at https://vis-www.cs.umass.edu/lfw/.

- InstructPix2Pix (IP2P) [2] is licensed under the MIT license and is available at https://github.com/timothybrooks/instruct-pix2pix.

- IP2P further builds upon stable-diffusion-v1-5 that is released under CreativeML- Open-RAIL-M License.

## C.4 Further information of the new dataset.

CelebA-CF and LFW-CF are based on real facial image datasets, such as CelebA and LFW, which include 40 attribute annotations. We note that because the original datasets have an imbalance between some pairs of two attributes, our datasets also possess a different skewness between the sensitive attribute and other attributes. For example, in CelebA and CelebA-CF, most images with the blond hair attribute are female, and most males do not have blond hair. We present the skewness values between the sensitive attribute and other non-sensitive attributes in CelebA-CF in Table C.1. Additionally, we displayed the group-specific failure rate identified through the filtering process in Table C.2.

Table C.1: **The skewness between the sensitive attribute and other non-sensitive attributes in CelebA-CF.**

| Attribute | Hair Length | Bangs | Wearing Hat | Brown Hair | Pale Skin | Big Lips | Mouth Slightly Open | Smiling | Wavy Hair |
|---|---|---|---|---|---|---|---|---|---|
| Skewness | 0.798 | 0.907 | 0.974 | 0.820 | 0.980 | 0.815 | 0.265 | 0.335 | 0.833 |

Table C.2: **The failure rate identified through the filtering process.** This table shows the proportion of images filtered out, calculated separately for each sensitive attribute (*e.g.*, female, male), in constructing the CelebA-CF and LFW-CF datasets. The sensitive attributes in the table represent the labels of the original images.

|  | CelebA-CF | LFW-CF |
|---|---|---|
| Female | 0.57 | 0.70 |
| Male | 0.80 | 0.69 |

# D    Implementation details

## D.1    Details on training datasets

For CelebA, we utilize the official train-validation-test split [25]. For LFW, we also use the official train-test split [13] and then divide the training data into a training and a validation set, with a ratio of 80:20.

CIFAR-10B is a modified dataset from CIFAR-10, as described in Section 5.2. We modify CIFAR-10 into a binary classification task by dividing the original 10 classes into two classes (classes 0-4 and 5-9). To introduce a fairness issue, we set the sensitive attribute $A$ as the presence of Gaussian noise and skew the dataset by randomly injecting the noise into 20% and 80% of the data in class 0 and class 1, respectively. Additionally, we introduce Contrast noise for the attribute $G$. Using the skew-ratio $\alpha$, we create a statistical correlation between $A$ and $G$ by adding the noise into $100 \times \alpha\%$ of the data samples with $A = 1$ and $100 \times (1 - \alpha)\%$ of the data samples with $A = 0$. Unless otherwise noted, we set $\alpha$ to 0.8. We partition the dataset into train-validation-test sets with a ratio of 64:16:20, respectively, maintaining consistent values for the two skewness ratios (*i.e.*, skewness between $A$ and $Y$, $G$ and $A$) across all sets during our experiments.

## D.2    Compute Infrastructure and optimization

Our all experiments including the dataset construction and performance comparison of existing methods and CKD were conducted using AMD Ryzen Threadripper PRO 3975WX CPUs and NVIDIA RTX A5000 GPUs. Our dataset generation was parallelized using 8 GPUs and took 2 days to complete. Training time for models used in the experiments for performance comparison ranges between 12 to 24 hours depending on the dataset and method used.

For CIFAR-10B, we use ResNet56 models with the Adam optimizer for 50 epochs. We set the mini-batch size and learning rate as 128 and 0.001, respectively. Because the skewness between $G$ and $A$ in CIFAR-10B test datasets varies, we compute the balanced CD over both the target class and the sensitive attribute for the consistent metric. For CelebA and LFW, we train ResNet18 models with the AdamW optimizer. We use the epoch size of 70 and 50 for each dataset, and set the mini-batch size, learning rate, and weight decay as 128, 0.001, and 1e-4, respectively. We use identical hyperparameters regarding the optimization for all methods. All results are averaged over results from four different random seeds.

## D.3    Implementation details of baselines and CKD

**CF-aware methods.** Scratch (+Aug) [7] minimizes the empirical cross-entropy loss computed using both original and counterfactual images. CP [36] has a regularization term that promotes the image pairs to be the same prediction. We use logits of a neural network model as representation vectors for the CP regularization term. Since Scratch (+aug) and CP utilize CTF samples, we generated these samples using IP2P with the same prompt in Appendix C.1 using the image-CFG of 7.5 and the Text-CFG of 2.0. We note that we do not apply any filtering process for their generated training datasets. SenSeI [42] uses two metrics for training: one for a pre-defined fair regularizer distance metric and the other obtained by fair metric learning. We use the same metrics as presented in their code. By generating the worst-case samples based on these metrics, we apply a fair regularization term to promote their predictions to be the same, as originally implemented. LASSI [31] minimizes an objective function which is composed of the classification loss, the reconstruction loss, and the adversarial loss to learn individually fair representation. We use the official code of LASSI as it is.

**GF-aware methods.** LBC [15] necessitates multiple full-training iterations, alternately re-weighting each group based on the given group fairness metric and re-training. Due to its high computation budget for iterative full-training, we limit the number of epochs for each training to 5 and repeat this process 14 times. COV [43] utilizes a fairness constraint based on the covariance between the group label and the signed distance of feature vectors from the decision boundary of a classifier. We minimize the constraint-regularized objective function through gradient descent optimization, instead of directly solving its optimization problem. MFD [16] employs an additional fairness-promoting regularization term based on Maximum Mean Discrepancy (MMD). For the MMD distance of the regularization term, we use the Gaussian RBF kernel with the variance parameter set as the mean of squared distance between all data points. We implemented SS [14] and RW [17] identically to the original algorithm.

**CF- and GF-aware methods.** The combinations of GF method and the augmentation were implemented so that GF methods train a model on their own objective function using training datasets augmented by generated CTF samples. The combinations of GF methods and CP optimize the objective functions of GF methods combined by the CP regularization. The CKD regularization term (5) builds upon representation vectors $f(\boldsymbol{\theta}, x)$. For this vector, we use the logits of a neural network model on LFW and CIFAR-10B. For CelebA, we utilize feature vectors from the penultimate layer of models as a representation vector since its training dataset is relatively much larger and more complex than others, leading to more fine-grained feature vectors.

Our code is available at `https://github.com/sumin-yu/CKD.git`.

## D.4 Hyperparameter search

The range of hyperparameter search used for Table 2 and Table 5 are shown in Table D.1. We utilize grid search to select hyperparameter values within a certain range. Note CKD and the combinations of GF-aware methods and CP have additional parameters $\lambda$ for CP loss. We use the same range as CP for $\lambda$. We also note that for LASSI, we only search the hyperparameter for an adversarial loss while maintaining other parameters as the same as used in their experiments on CelebA.

Table D.1: Hyperparameters and search ranges for each method.

| Method | Hyperparameter | Search range |
|---|---|---|
| CP [36] | CP strength $\lambda$ | $[10^{-2}, 10^2]$ |
| SenSeI [42] | Fair regularization strength $\rho$ | $[10^{-2}, 10^2]$ |
| LASSI [31] | Adversarial loss weight $\lambda_2$ | $[10^{-3}, 10^{-1}]$ |
| COV [43] | Covariance strength $\lambda$ | $[10^{-2}, 10^2]$ |
| MFD [16] | MMD strength $\lambda$ | $[10^{-1}, 10^6]$ |
| LBC [15] | LR for re-weights $\eta$ | $[10^{-1}, 10^3]$ |
| CKD | CKD strength $\mu$ | $[10^{-1}, 10^3]$ |

# E  Rate of Flipped Predictions (RFP)

## E.1  RFP measurement on CelebA

To measure RFP on CelebA, we assume that the hair length of facial images is $G$, *i.e.*, it is correlated with, but not caused by, the sensitive attribute. Then, we use Stochastic Differential Editing (SDEdit) [28], an image editing method based on a diffusion model, to modify the hair length in each image. SDEdit selectively edits specific regions of a given image based on the colored stroke. Namely, SDEdit depicts the image region indicated by the stroke with the given color in the most plausible manner. By doing so, SDEdit generates realistic and faithful edited images, while preventing changes in the region not indicated by the strokes. To utilize SDEdit, we randomly select 40 samples for each group of the same target label and sensitive attribute from original samples of CelebA-CTF pairs and then we manually apply strokes on the hair of facial images for a total of 160 samples. Specifically, To extend the hair length, we applied strokes with the hair color to the areas where the hair should grow. Conversely, to shorten the hair length, we applied strokes with the background color to the areas where the hair should be removed. After this process, we utilize the official PyTorch implementation of Meng et al. [28] to edit images with the applied strokes. Figure E.1 shows some examples of images edited by SDEdit.

Table E.1: The skewness between the sensitive attribute and other attributes, as well as the accuracy for each attribute after re-training a linear classifier on the top of the CP-trained model.

| | Hair Length | Bangs | Wearing Hat | Pale Skin | Mouth Slightly Open |
|---|---|---|---|---|---|
| Acc (%) | 64.93 | 68.07 | 72.63 | 50.79 | 56.79 |
| Skewness | 0.8 | 0.91 | 0.97 | 0.98 | 0.27 |

**Original images**

**Edited images (change the hair length, keeping the others)**

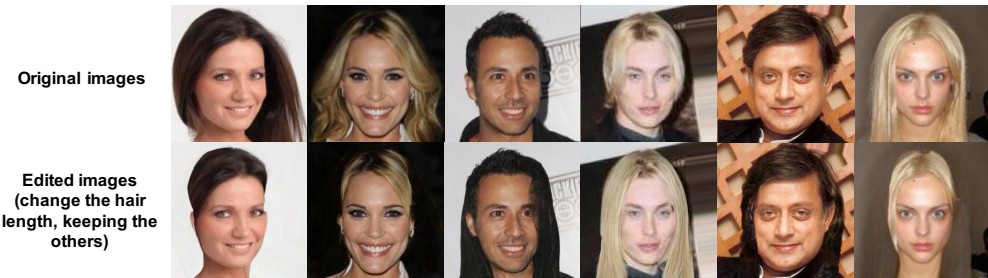

Figure E.1: **Examples of CelebA measuring RFP with respect to "hair length".** If the person in the original image (first row) had long hair, we create a modified image (second row) with shorter hair and vice versa.

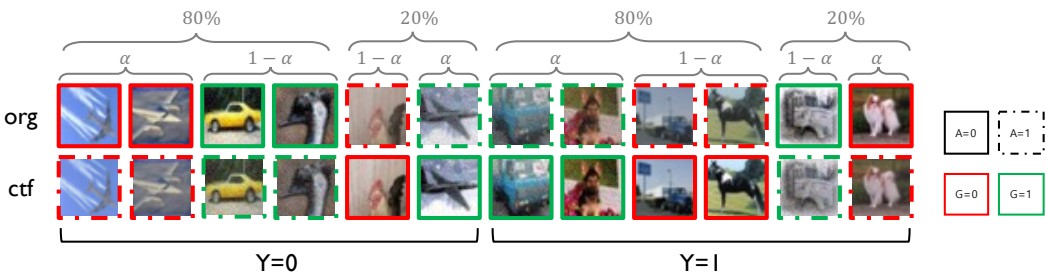

Figure E.2: **Illustration of CIFAR-10B.** The sensitive attribute $A$ is characterized by the line type ($A = 0$ for a solid line, $A = 1$ for a dashed line), while another attribute $G$ is denoted by the color of the line ($G = 0$ for a red line, $G = 1$ for a green line).

## E.2 Discussion about selecting $G$ on CelebA

As mentioned in Section 5.2, we intuitively chose "hair length" as $G$ on CelebA since $G$ is highly correlated with, but not caused by, the sensitive attribute $A$. However, we introduce a more generalized approach for choosing $G$ by leveraging a CP-trained model that exhibits low CD but high DEO. Specifically, using a pre-defined set of attributes, we first train a linear classifier on the top of the feature extractor from the CP-trained model for each attribute. In cases where annotations are not available, CLIP-based pseudo labels can be utilized. Based on the accuracy of each linear classifier, we can then identify which attributes the CP-trained model learns more, indicating potential heavy reliance on these attributes. Finally, we can select $G$ attributes based on two criteria: (1) high accuracy of a linear classifier and (2) high correlation (not causation) with the sensitive attribute.

To validate this approach, we conducted an experiment on CelebA dataset using a subset of 40 pre-annotated attributes and the "hair length" ("hair length" labels are predicted by CLIP as it is not originally labeled in CelebA). Table E.1 displays the accuracy for each attribute after training a linear classifier on the top of the CP-trained model, alongside the skewness between the sensitive attribute and other attributes. Attributes like "Pale Skin" show a high correlation with the sensitive attribute but low accuracies, suggesting CP might not rely on them (not satisfying the second condition). "Mouth Sightly Open" exhibits low correlation and accuracy, thus not being considered as $G$ (failing the first condition). In contrast, attributes such as "bangs", "wearing hat", and "hair length" exhibit both high correlation values and accuracies, indicating that they are promising candidates for the attribute $G$.

# F Further discussion about the faithfulness assumption

The faithful assumption states that if some two variables are statistically independent, they are d-separated, *i.e.*, there is no connected path between them. Thus, under this assumption, GF-aware methods, which enforce independence between the sensitive attribute and the target label, can achieve CF simultaneously when they successfully achieve GF. The previous works [1, 35] provide the same argument that GF implies CF under the faithfulness assumption. However, some other previous works [18] have demonstrated that while the faithfulness assumption is crucial for causal inference literature, it may not always hold true, especially in complex real-world scenarios. Moreover, the result for GF-aware methods in Table 5 reveals that the GF-aware methods have a minimal impact on improving CD, implying that the faithfulness assumption does not hold in CelebA and LFW datasets.

# G Additional results

## G.1 Result tables with standard deviation

We report the standard deviation values of the performance comparison results in Table G.1 for CIFAR-10B, CelebA, and LFW respectively. The standard deviation values are calculated over four different seeds.

## G.2 Impact of the robustness to $G$ of the teacher model on CKD with CelebA and LFW

We analyze the effectiveness of the teacher model on real image datasets. We report the performance of CKD using variants of teacher models that are more or less robust to $G$ on CelebA and LFW. We consider three teacher models, ordered by robustness to $G$: CP ($\boldsymbol{\theta}_{CP}^{T}$), Scratch ($\boldsymbol{\theta}_{Scratch}^{T}$), and CKD model with a Scratch teacher ($\boldsymbol{\theta}_{CKD}^{T}$). Table G.2 displays ACC, DEO, CD (on CelebA and LFW), and RFP (on CelebA) depending on the teacher models. Since we generate a dataset for RFP measurements on CelebA with "hair length" as $G$, we report RFP values only for CelebA. Through the result, we observe that compared to vanilla-trained teachers $\boldsymbol{\theta}_{Scratch}^{T}$, using more robust teachers (*e.g.*, $\boldsymbol{\theta}_{CKD}^{T}$) achieves slightly better or competitive DEO and CD, while employing less robust teachers (*e.g.*, $\boldsymbol{\theta}_{CP}^{T}$) significantly degrades DEO, which are consistent with the results in Section 5.3 on CIFAR-10B.

## G.3 Analaysis on CKD

**Ablation study.** To study the effectiveness of our CKD regularization term, we additionally consider a method that is a naive combination of a typical KD method proposed by Hinton et al. [12] and CP [36] (*i.e.*, HKD+CP). Note that this combination can be considered as a baseline method that considers both CF and GF if it uses a robust teacher model to $G$, because the method robustifies the model with respect to $G$ while achieving CF. Table G.5 compares the method with our CKD. As we expected, the results show that HKD+CP improves both DEO and CD simultaneously. However, its performance is still suboptimal compared to CKD, showing CKD is more effective than the naive combination of KD and CP.

**CKD on feature vectors vs logits.** CKD can utilize either feature or logit vectors as the target vectors, i.e., $f(\theta, x)$. For CIFAR-10B and LFW, where we use logits as the target vectors in our experiments, we displayed the performance of CKD using feature vectors as the target vectors in Table G.6. The results demonstrate that CKD using feature vectors exhibits comparable performance to those with logits. Moreover, we can get even better performance on CIFAR-10B using feature vectors, demonstrating that CF and GF can be achieved simultaneously regardless of which type of target vectors is used.

**Sensitiveness of $\mu$.** $\mu$ is the regularization strength for the CKD loss term. Specifically, as $\mu$ increases, we expect improvements in both DEO and CD. Table G.7 shows the performance of CKD across different values of $\mu$, aligning with our expectations. Additionally, we note that CKD performance is insensitive to $\mu$.

**The implication of using non-curated IP2P.** Uncurated CTF datasets are imperfect. Specifically, some samples generated from the original images in our test datasets were filtered out because the images either showed minimal changes or had alterations that affected non-sensitive attributes including the target attributes. Consequently, the more such incomplete samples exist, the more they will negatively impact the performance of our method. To assess how sensitive CKD is to incomplete CTF training samples, we conducted additional experiments on CIFAR-10B by varying the ratio of incomplete CTF samples in the training set. For a given ratio $\alpha$, we assumed that half of the incomplete samples are nearly unchanged, while the other half are samples where both the target and sensitive attributes are altered. We varied $\alpha$ from 20% to 60% in 10% increments and reported the accuracy, CD, and DEO of CKD in the table below. The results in Table G.3 indicate that CKD significantly improves both CD and DEO compared to Scratch, even for high $\alpha$ s. Although this phenomenon has not been fully explained, we hypothesize that the robustness can be attributed to the distillation process, as empirically demonstrated in [8].

### G.4 Additional experimental results on counterfactual samples.

We additionally report the accuracy (acc-CTF) and DEO (DEO-CTF) for CKD and several baseline methods on counterfactual samples in CelebA-CF in Table G.4.

### G.5 Additional experimental results for other metrics proposed by [33].

We first note that the Counterfactual Disparity (CD) we used is the same metric as the Switch Rate (SR) proposed by [33]. We computed P2NR (another metric proposed by [33]) on CelebA and obtained values of 0.036, 0.165, and 0.339 for Scratch, CP, and CKD, respectively. These results indicate that CKD achieves low CD with a balanced rate of misclassification across the labels. Additionally, we would like to emphasize that Pinto et al. focused on scenarios where GF does not imply CF in their experiments—highlighting cases where the faithfulness assumption, which can be overly stringent, does not hold (see line 323 in their paper). However, our work primarily explores the converse: whether CF can imply GF depending on the presence of $G$, independent of the faithfulness assumption.

Table G.1: **Evaluation of GF and CF of fair-training for image classification**. We put the results from Table 2 and Table 5 together with the standard deviation values over four different seeds.

(a) Standard deviations on CIFAR-10B.

| Method | CIFAR-10B ($\alpha$=0.8) | | |
| --- | --- | --- | --- |
| | Acc ↑ | CD ↓ | DEO ↓ |
| Scratch | $78.01_{\pm 0.85}$ | $17.90_{\pm 2.43}$ | $27.46_{\pm 2.65}$ |
| CF-aware training | | | |
| Scratch+aug [7] | $75.38_{\pm 0.79}$ | $9.33_{\pm 0.88}$ | $15.25_{\pm 1.19}$ |
| CP [36] | $75.26_{\pm 1.87}$ | $2.05_{\pm 0.15}$ | $33.23_{\pm 7.56}$ |
| SenSeI [42] | $77.21_{\pm 0.83}$ | $16.32_{\pm 1.23}$ | $24.18_{\pm 1.70}$ |
| GF-aware training | | | |
| SS [14] | $74.77_{\pm 0.41}$ | $16.42_{\pm 1.34}$ | $25.73_{\pm 2.12}$ |
| RW [17] | $76.53_{\pm 0.57}$ | $12.15_{\pm 1.23}$ | $18.94_{\pm 1.71}$ |
| COV [43] | $79.03_{\pm 0.49}$ | $13.90_{\pm 0.39}$ | $24.05_{\pm 1.09}$ |
| MFD [16] | $76.84_{\pm 0.92}$ | $12.24_{\pm 1.51}$ | $15.39_{\pm 1.28}$ |
| LBC [15] | $76.16_{\pm 1.36}$ | $15.01_{\pm 2.26}$ | $17.12_{\pm 4.68}$ |
| both CF and GF-aware training | | | |
| SS+aug | $73.45_{\pm 0.46}$ | $9.95_{\pm 0.58}$ | $15.21_{\pm 2.06}$ |
| RW+aug | $76.15_{\pm 0.52}$ | $12.93_{\pm 0.96}$ | $20.94_{\pm 2.57}$ |
| COV+aug | $76.52_{\pm 0.56}$ | $8.17_{\pm 0.22}$ | $15.04_{\pm 0.96}$ |
| MFD+aug | $77.10_{\pm 0.58}$ | $11.16_{\pm 1.08}$ | $14.79_{\pm 2.15}$ |
| LBC+aug | $75.82_{\pm 0.54}$ | $9.01_{\pm 0.67}$ | $15.29_{\pm 0.56}$ |
| SS+CP | $76.54_{\pm 1.48}$ | $3.14_{\pm 0.26}$ | $9.08_{\pm 1.22}$ |
| RW+CP | $75.68_{\pm 0.37}$ | $8.83_{\pm 0.66}$ | $13.92_{\pm 0.94}$ |
| COV+CP | $77.74_{\pm 0.52}$ | $4.30_{\pm 0.33}$ | $19.42_{\pm 1.74}$ |
| MFD+CP | $76.67_{\pm 1.00}$ | $10.01_{\pm 0.57}$ | $13.17_{\pm 0.95}$ |
| LBC+CP | $76.88_{\pm 2.64}$ | $3.02_{\pm 1.29}$ | $12.45_{\pm 1.93}$ |
| CKD ($\lambda = 0$) | $76.32_{\pm 0.59}$ | $8.59_{\pm 1.32}$ | $11.23_{\pm 1.04}$ |
| CKD | $78.49_{\pm 0.66}$ | $2.85_{\pm 0.20}$ | $7.30_{\pm 0.46}$ |

(b) Standard deviations on CelebA and LFW.

| Method | CelebA (and CelebA-CF) | | | LFW (and LFW-CF) | | |
| --- | --- | --- | --- | --- | --- | --- |
| | Acc ↑ | CD ↓ | DEO ↓ | Acc ↑ | CD ↓ | DEO ↓ |
| Scratch | $95.53_{\pm 0.06}$ | $10.26_{\pm 1.33}$ | $47.10_{\pm 5.57}$ | $90.85_{\pm 0.27}$ | $18.06_{\pm 1.89}$ | $7.66_{\pm 0.49}$ |
| CF-aware training | | | | | | |
| Scratch+aug [7] | $95.41_{\pm 0.15}$ | $4.65_{\pm 0.86}$ | $44.71_{\pm 2.57}$ | $90.34_{\pm 0.58}$ | $12.15_{\pm 1.29}$ | $7.86_{\pm 1.70}$ |
| CP [36] | $94.10_{\pm 0.08}$ | $2.53_{\pm 1.26}$ | $51.01_{\pm 1.71}$ | $89.77_{\pm 0.90}$ | $9.20_{\pm 0.56}$ | $8.74_{\pm 1.43}$ |
| SenSeI [42] | $95.33_{\pm 0.30}$ | $8.00_{\pm 0.59}$ | $52.32_{\pm 5.26}$ | $87.75_{\pm 3.78}$ | $16.09_{\pm 3.70}$ | $9.23_{\pm 1.38}$ |
| LASSI [31] | $91.07_{\pm 0.27}$ | $9.69_{\pm 0.78}$ | $31.79_{\pm 3.17}$ | - | - | - |
| GF-aware training | | | | | | |
| SS [14] | $95.44_{\pm 0.09}$ | $9.13_{\pm 2.73}$ | $42.95_{\pm 3.87}$ | $90.43_{\pm 0.21}$ | $18.19_{\pm 1.27}$ | $6.75_{\pm 0.33}$ |
| RW [17] | $95.16_{\pm 0.08}$ | $5.50_{\pm 0.46}$ | $24.21_{\pm 1.76}$ | $90.87_{\pm 0.25}$ | $18.68_{\pm 2.91}$ | $6.92_{\pm 0.96}$ |
| COV [43] | $94.42_{\pm 0.16}$ | $7.72_{\pm 2.89}$ | $34.04_{\pm 4.43}$ | $90.85_{\pm 0.50}$ | $16.43_{\pm 2.12}$ | $6.99_{\pm 1.23}$ |
| MFD [16] | $94.37_{\pm 0.77}$ | $4.61_{\pm 1.77}$ | $19.00_{\pm 6.44}$ | $90.47_{\pm 0.10}$ | $16.07_{\pm 2.01}$ | $2.15_{\pm 0.51}$ |
| LBC [15] | $94.92_{\pm 0.28}$ | $6.24_{\pm 0.69}$ | $22.61_{\pm 1.79}$ | $90.71_{\pm 0.61}$ | $15.76_{\pm 0.86}$ | $3.56_{\pm 2.02}$ |
| both CF and GF-aware training | | | | | | |
| SS+aug | $95.17_{\pm 0.02}$ | $5.24_{\pm 1.02}$ | $40.80_{\pm 2.86}$ | $89.96_{\pm 0.24}$ | $15.23_{\pm 2.21}$ | $6.82_{\pm 0.88}$ |
| RW+aug | $95.13_{\pm 0.04}$ | $5.34_{\pm 0.59}$ | $24.63_{\pm 1.58}$ | $90.76_{\pm 0.16}$ | $18.63_{\pm 2.07}$ | $6.71_{\pm 1.38}$ |
| COV+aug | $94.08_{\pm 0.41}$ | $8.11_{\pm 2.26}$ | $29.03_{\pm 0.72}$ | $90.47_{\pm 0.20}$ | $13.65_{\pm 1.71}$ | $6.78_{\pm 0.09}$ |
| MFD+aug | $93.78_{\pm 0.80}$ | $3.87_{\pm 0.84}$ | $14.36_{\pm 4.39}$ | $89.90_{\pm 0.62}$ | $19.36_{\pm 2.41}$ | $2.47_{\pm 0.75}$ |
| LBC+aug | $94.39_{\pm 1.44}$ | $9.32_{\pm 4.20}$ | $36.08_{\pm 11.36}$ | $88.66_{\pm 1.25}$ | $12.41_{\pm 2.02}$ | $2.79_{\pm 1.36}$ |
| SS+CP | $94.54_{\pm 0.09}$ | $2.40_{\pm 0.35}$ | $37.97_{\pm 2.27}$ | $88.70_{\pm 0.82}$ | $6.13_{\pm 1.17}$ | $4.26_{\pm 1.74}$ |
| RW+CP | $95.19_{\pm 0.13}$ | $4.67_{\pm 0.76}$ | $25.56_{\pm 2.87}$ | $90.87_{\pm 0.28}$ | $15.24_{\pm 1.67}$ | $6.16_{\pm 0.19}$ |
| COV+CP | $94.29_{\pm 0.18}$ | $5.36_{\pm 1.00}$ | $51.63_{\pm 0.67}$ | $91.23_{\pm 0.37}$ | $11.91_{\pm 2.18}$ | $6.52_{\pm 1.05}$ |
| MFD+CP | $93.81_{\pm 0.30}$ | $3.47_{\pm 0.52}$ | $23.31_{\pm 0.74}$ | $89.39_{\pm 1.90}$ | $15.15_{\pm 1.24}$ | $1.90_{\pm 1.03}$ |
| LBC+CP | $95.12_{\pm 0.10}$ | $4.72_{\pm 0.87}$ | $22.78_{\pm 2.26}$ | $89.92_{\pm 0.28}$ | $8.33_{\pm 1.07}$ | $3.02_{\pm 0.58}$ |
| CKD ($\lambda = 0$) | $94.12_{\pm 0.23}$ | $4.31_{\pm 1.47}$ | $14.11_{\pm 1.25}$ | $90.76_{\pm 0.13}$ | $12.42_{\pm 2.69}$ | $2.64_{\pm 0.14}$ |
| CKD | $93.08_{\pm 0.46}$ | $4.44_{\pm 0.70}$ | $13.23_{\pm 1.30}$ | $89.26_{\pm 0.45}$ | $7.94_{\pm 0.89}$ | $1.88_{\pm 0.67}$ |

Table G.2: **Impact of robustness to $G$ of the teacher model on CKD with Celeb and LFW.** $\theta_{\text{CKD}}^T$ and $\theta_{\text{CP}}^T$ are CKD and CP trained teacher models. $\theta_{\text{Scratch}}^T$ is a vanilla-trained teacher model. RFP$^T$ denotes how a teacher is biased towards $G$. DEO, CD, RFP are metrics for evaluating GF, CF, and bias towards $G$, respectively. Since we generate a dataset for RFP measurements on CelebA with "hair length" as $G$, we report RFP values only for CelebA.

| Method | | CelebA | | | | | LFW | |
| | RFP$^T \downarrow$ | Acc $\uparrow$ | DEO $\downarrow$ | CD $\downarrow$ | RFP $\downarrow$ | Acc $\uparrow$ | DEO $\downarrow$ | CD $\downarrow$ |
|---|---|---|---|---|---|---|---|---|
| CKD w/ $\theta_{\text{Scratch}}^T$ | 15.27 | 93.08 | 13.23 | 4.44 | 10.85 | 89.26 | 1.88 | 7.94 |
| CKD w/ $\theta_{\text{CKD}}^T$ | 10.85 | 93.98 | 14.37 | 4.05 | 11.64 | 89.17 | 1.48 | 8.07 |
| CKD w/ $\theta_{\text{CP}}^T$ | 20.37 | 94.25 | 34.49 | 3.28 | 16.61 | 89.85 | 9.26 | 8.32 |

Table G.3: **The implication of using non-curated IP2P.**

| | Acc $\uparrow$ | CD $\downarrow$ | DEO $\downarrow$ |
|---|---|---|---|
| Scratch | 78.01 | 17.90 | 27.46 |
| CKD | 78.49 | 2.85 | 7.30 |
| CKD (20%) | 79.82 | 2.77 | 11.41 |
| CKD (30%) | 79.76 | 2.88 | 12.78 |
| CKD (40%) | 79.70 | 2.94 | 12.88 |
| CKD (50%) | 79.72 | 2.85 | 14.10 |
| CKD (60%) | 79.61 | 3.01 | 14.94 |

Table G.4: **The accuracy and DEO on counterfactual samples in CelebA-CF.**

| | Acc-CTF $\uparrow$ | DEO-CTF $\downarrow$ |
|---|---|---|
| Scratch | 77.22 | 15.92 |
| SS | 81.43 | 28.74 |
| RW | 79.75 | 23.61 |
| LBC | 78.06 | 33.95 |
| CP | 75.21 | 62.82 |
| CKD | 90.08 | 21.86 |

Table G.5: **Evaluation of group fairness (GF) and counterfactual fairness (CF) of fair-training for image classification**. The details are the same as Table 5. "HKD+CP" denotes a model that naively combines Knowledge Distillation [12] with CP [36].

| Method | CIFAR-10B ($\alpha$=0.8) | | | CelebA (and CelebA-CF) | | | LFW (and LFW-CF) | | |
| | Acc $\uparrow$ | DEO $\downarrow$ | CD $\downarrow$ | Acc $\uparrow$ | DEO $\downarrow$ | CD $\downarrow$ | Acc $\uparrow$ | DEO $\downarrow$ | CD $\downarrow$ |
|---|---|---|---|---|---|---|---|---|---|
| Scratch | 78.36 | 27.28 | 17.68 | 95.53 | 47.10 | 10.36 | 90.85 | 7.66 | 18.06 |
| HKD+CP | 79.18 | 16.54 | 2.40 | 93.95 | 33.98 | 4.40 | 89.11 | 3.76 | 8.78 |
| CKD ($\lambda = 0$) | 75.63 | 6.33 | 8.94 | 94.12 | 14.11 | 4.31 | 90.76 | 2.64 | 12.47 |
| CKD | 78.46 | 7.11 | 2.86 | 93.08 | 13.23 | 4.44 | 89.26 | 1.88 | 7.94 |

Table G.6: **CKD on feature vectors vs logits.**

| Method | CIFAR-10B ($\alpha$=0.8) | | | LFW | | |
| | Acc $\uparrow$ | DEO $\downarrow$ | CD $\downarrow$ | Acc $\uparrow$ | DEO $\downarrow$ | CD $\downarrow$ |
|---|---|---|---|---|---|---|
| CKD w/ logit | 78.46 | 7.11 | 2.86 | 89.26 | 1.88 | 7.94 |
| CKD w/ feature | 78.24 | 2.64 | 1.23 | 88.37 | 3.98 | 6.22 |

Table G.7: **Sensitivity of** $\mu$.

| | CelebA | | | | LFW | | |
|---|---|---|---|---|---|---|---|
| $\mu$ | Acc $\uparrow$ | DEO $\downarrow$ | CD $\downarrow$ | $\mu$ | Acc $\uparrow$ | DEO $\downarrow$ | CD $\downarrow$ |
| 0.01 | 94.13 | 14.83 | 4.4 | 0.01 | 90.05 | 2.82 | 15.01 |
| 0.1 | 94.13 | 14.45 | 3.03 | 0.1 | 90.09 | 2.39 | 14.19 |
| 1.0 | 93.63 | 13.84 | 4.24 | 1.0 | 89.78 | 2.75 | 13.66 |
| 7.0 | 93.08 | 13.23 | 4.44 | 10.0 | 89.23 | 1.85 | 13.82 |
| 10.0 | 92.93 | 13.67 | 4.10 | 50.0 | 88.83 | 1.8 | 7.80 |

# H  Datasheet for dataset

## H.1  Motivation

**For what purpose was the dataset created?** These datasets were created for evaluating counterfactual fairness in image classifiers. Furthermore, since our datasets contain counterfactual images generated from real-world images, our datasets can be also used for analyzing the relationship between counterfactual and group fairness on image datasets. For more discussion of the motivation behind our datasets, see Section 1.

**Who created the dataset (e.g., which team, research group) and on behalf of which entity (e.g., company, institution, organization)?** The datasets were created by the authors of this paper who were affiliated with Seoul National University and NAVER AI LAB.

**Who funded the creation of the dataset?** Funding was provided by the National Research Foundation of Korea (NRF); Institute of Information & Communications Technology Planning & Evaluation (IITP); and the SNU-Naver Hyperscale AI Center.

## H.2  Composition

**What do the instances that comprise the dataset represent (e.g., documents, photos, people, countries)?** The instances represent synthetically generated images and corresponding real-world original images from two popular benchmark facial image datasets, CelebA [25] and LFW [13].

**How many instances are there in total (of each type, if appropriate)?** CelebA-CF and LFW-CF contain a total of 230 and 144 image pairs of original and counterfactual images, respectively.

**Does the dataset contain all possible instances or is it a sample (not necessarily random) of instances from a larger set?** We uniformly sampled a subset of test images in CelebA and LFW to balance the target and group labels (see more details in Section 2). Then, we made our datasets including all possible samples according to our filtering process.

**What data does each instance consist of?** Each instance contains a pair of original and counterfactual images.

**Is there a label or target associated with each instance?** Yes, there are 40 binary annotations that originated from CelebA and LFW.

**Is any information missing from individual instances?** No

**Are relationships between individual instances made explicit (e.g., users' movie ratings, social network links)?** Yes, instances that correspond to a counterfactual pair are explicitly annotated as such in our dataset. Otherwise, there are no relationships between individual instances.

**Are there recommended data splits (e.g., training, development/validation, testing)?** No, the dataset is created for the purpose of testing.

**Are there any errors, sources of noise, or redundancies in the dataset?** Using IP2P as the initial step in constructing our datasets might introduce some noise or errors in the datasets. Refer to Appendix B for further details.

**Is the dataset self-contained, or does it link to or otherwise rely on external resources (e.g., websites, tweets, other datasets)?** Yes, it is self-contained.

**Does the dataset contain data that might be considered confidential (e.g., data that is protected by legal privilege or by doctor patient confidentiality, data that includes the content of individuals' non-public communications)?** No

**Does the dataset contain data that, if viewed directly, might be offensive, insulting, threatening, or might otherwise cause anxiety?** Yes, the dataset may cause some anxiety about sex labels. See Section 2 and Appendix B.

**Does the dataset identify any subpopulations (e.g., by age, gender)?** Yes, our datasets were created after evaluating whether counterfactual samples regarding visually perceived sexual traits were generated correctly or not. This evaluation was conducted by five human annotators. Thus, our datasets contain the identification of visually perceived sexual traits which represent some statistically representative features for each sex. See more discussion in Section 2.

**Is it possible to identify individuals (i.e., one or more natural persons), either directly or indirectly (i.e., in combination with other data) from the dataset?** Yes, our datasets are generated from CelebA and LFW, which are facial datasets collected on the internet.

**Does the dataset contain data that might be considered sensitive in any way (e.g., data that reveals race or ethnic origins, sexual orientations, religious beliefs, political opinions or union memberships, or locations;**

**financial or health data; biometric or genetic data; forms of government identification, such as social security numbers; criminal history)?** Yes, we set sex as the sensitive attribute and created our CTF samples with the sensitive attribute flipped.

## H.3   Collection Process

**How was the data associated with each instance acquired?** Our datasets are generated through image editing using IP2P (See Section 2).

**What mechanisms or procedures were used to collect the data (e.g., hardware apparatuses or sensors, manual human curation, software programs, software APIs)?** Refer to Section 2 for a complete description of our data generation process.

**If the dataset is a sample from a larger set, what was the sampling strategy (e.g., deterministic, probabilistic with specific sampling probabilities)?** Original test samples in our datasets were uniformly sampled from the test datasets of CelebA and LFW.

**Who was involved in the data collection process (e.g., students, crowdworkers, contractors) and how were they compensated (e.g., how much were crowdworkers paid)?** The filtering process for our datasets involved five student annotators who received about 18 USD per hour for their wage.

**Over what timeframe was the data collected?** Our datasets were generated and evaluated over one month.

**Were any ethical review processes conducted (e.g., by an institutional review board)?** No

**Did you collect the data from the individuals in question directly, or obtain it via third parties or other sources (e.g., websites)?** No, we initially obtained the data from publicly available sources. Subsequently, we edited the data and filtered the edited one through human annotators.

**Were the individuals in question notified about the data collection?** Not applicable

**Did the individuals in question consent to the collection and use of their data?** Not applicable

**If consent was obtained, were the consenting individuals provided with a mechanism to revoke their consent in the future or for certain uses?** Not applicable

**Has an analysis of the potential impact of the dataset and its use on data subjects (e.g., a data protection impact analysis) been conducted?** Not applicable

## H.4   Preprocessing/cleaning/labeling

**Was any preprocessing/cleaning/labeling of the data done (e.g., discretization or bucketing, tokenization, part-of-speech tagging, SIFT feature extraction, removal of instances, processing of missing values)?** Yes, we filtered our generated datasets with human annotators. Refer to Section 2 for a complete description of our filtering process.

**Was the "raw" data saved in addition to the preprocessed/cleaned/labeled data (e.g., to support unanticipated future uses)?** No, however, raw data can be reproduced by applying IP2P as described in Section 2.

**Is the software that was used to preprocess/clean/label the data available?** Yes, refer to the Section 2.

## H.5   Uses

**Has the dataset been used for any tasks already?** Yes, we applied our datasets to evaluate CF in image classifiers in Section 3 and 6 and analyze the relationship between CF and GF in Section 5.

**Is there a repository that links to any or all papers or systems that use the dataset?** We will provide a link to a repository on GitHub that includes references to all papers and systems utilizing the dataset.

**What (other) tasks could the dataset be used for?** There is no other task where our dataset can be used. The dataset is exclusively designed for evaluating counterfactual fairness in real-world image datasets.

**Is there anything about the composition of the dataset or the way it was collected and preprocessed/cleaned/labeled that might impact future uses?** Because our datasets were generated through the image editing technique, IP2P [2], they may contain implicit biases or errors, which are present in the IP2P model [27**?** ]. While we have conducted a thorough human filtering and validation process to minimize these issues in our dataset, future users should still be aware of these limitations.

**Are there tasks for which the dataset should not be used?** The dataset should not be employed for tasks where the limitations discussed in Appendix B could pose critical issues, or for tasks that are not for research purposes.

### H.6 Distribution

**Will the dataset be distributed to third parties outside of the entity (e.g., company, institution, organization) on behalf of which the dataset was created?** Yes, the datasets will be made publicly available.

**How will the dataset will be distributed (e.g., tarball on website, API, GitHub)?** The dataset will be distributed using tarball on the website. Refer to Appendix C.

**When will the dataset be distributed?** The datasets will be made publicly available upon acceptance.

**Will the dataset be distributed under a copyright or other intellectual property (IP) license, and/or under applicable terms of use (ToU)?** The datasets will be distributed under the CC BY 4.0 license.

**Have any third parties imposed IP-based or other restrictions on the data associated with the instances?** No

**Do any export controls or other regulatory restrictions apply to the dataset or to individual instances?** No

### H.7 Maintenance

**Who will be supporting/hosting/maintaining the dataset?** The datasets are hosted, supported, and maintained by the authors.

**How can the owner/curator/manager of the dataset be contacted (e.g., email address)?** The corresponding author can be contacted by the e-mail address which will be listed on the first page of this paper after camera-ready.

**Is there an erratum?** No

**Will the dataset be updated (e.g., to correct labeling errors, add new instances, delete instances)?** No future updates are currently planned. However, we will monitor the GitHub repository for related issues and address any problems that arise.

**If the dataset relates to people, are there applicable limits on the retention of the data associated with the instances (e.g., were the individuals in question told that their data would be retained for a fixed period of time and then deleted)?** Not applicable

**Will older versions of the dataset continue to be supported/hosted/maintained?** Yes, if the datasets are updated, we will maintain the older versions.

**If others want to extend/augment/build on/contribute to the dataset, is there a mechanism for them to do so?** Yes, we make our code and datasets public, and hence others can contribute or extend to our work and datasets.