# OpenReview forum: "Do Counterfactually Fair Image Classifiers Satisfy Group Fairness? -- A Theoretical and Empirical Study"
_NeurIPS.cc/2024/Datasets_and_Benchmarks_Track — NeurIPS 2024 Track Datasets and Benchmarks Poster_

### Official Review · Reviewer_sLrQ · 2024-07-22

**Rating:** 5
**Confidence:** 3

**Review:**

The work contains useful insights and contributions in a variety of directions, including the complimentary counterfactual versions of existing datasets, experimental investigation of group- and counterfactual-fairness with these datasets, theoretical discussion of the relationship of these two groups, and additional experimentation with CIFAR-10. The work is strengthened by the fact that the authors are building on existing datasets. The contributions are interesting and generally cohesive.

[Concerns] However, I have several concerns about the work.
* The impact of the new dataset is somewhat limited (and only used for a subset of the experiments), and there is relatively little discussion of details about the new dataset.
* In addition, while the theoretical framework and CIFAR-10 experiments are insightful towards the overall study, they don't especially relate to the venue's focus on datasets and benchmarking. In addition, there exist similar datasets, e.g. https://ojs.aaai.org/index.php/AAAI/article/view/29333 and the unique value of the proposed datasets over the previous works is not clear
* Furthermore, nuances around the visual representation of gender are lacking, and especially how subjectivity in perceived gender is operationalized in the creation of the dataset and visual-attribute (hair) related tasks.
* Finally, there are some clarity / readability issues: the use of so many acronyms can make readability challenging at times and there are several sections where additional details / analyses would be useful.

**Strengths:**

* This work usefully introduces counterfactual versions of existing datasets, unlocking more details and thorough analyses of models.
* The data construction process includes additional validation of in-painting methods via human review on two axes of analysis.
* The suite of analyses and experimentation related to counterfactual and group fairness that is informed by  the counterfactual datasets and leads towards useful theoretical insights.
* The authors connect the dataset to interesting theoretical results and validate their findings with additional experimentation using CIFAR-10.

**Additional Feedback:**

My primary concern is the limited relevance of much of the theoretical / experimental contributions of this work to the dataset + benchmarks track and the missing details about the counterfactual dataset itself.

**Clarity:**

Generally the paper is clear, although the large set of acronyms makes readability challenging and the paper feels sparse in some details (please see last bullet of "Concerns" section of "Review").

**Correctness:**

In general, the claims made seem reasonable, although additional dataset details discussed in previous sections would be useful.

**Documentation:**

Yes, there is good discussion of the data collection process, especially regarding the human review process. This section would be strengthened by inclusion of the 8 example images provided to annotators (L.547) so that users have a better idea of how ground truth examples may have shaped annotator perceptions of the subjective attributes.
In addition, the details about ethical considerations included in the supplementary section are useful – it would be helpful to include more detail about what is meant by "they should not utilize our datasets for gender categorization", i.e. whether users may not train gender classification models with the data.

**Ethics:**

Additional discussion about the representation of gender and correlation with target attributes such as blonde and hair length would be useful. This relates to the "Data quality and representativeness" and "Discrimination, bias, and fairness" flags.

**Limitations:**

Please see "Concerns" section of Review.

**Opportunities For Improvement:**

I discuss here opportunities for improvement pertaining to each of the numbered "Concerns" from the Review section.

* The work would be strengthend with the addition of details about the original CelebA and LFW datasets including composition and possible biases. In addition, there could be additional details, such as the rate of different hair lengths, group-specific failure rates identified via the annotations, and rate of co-occurring attributes like age and skin color.
* Contextualize the work with the related https://ojs.aaai.org/index.php/AAAI/article/view/29333, especially how the proposed dataset provides additional insights. To increase relevance to this venue, include details about how the theoretical and empirical analyses may lend to broader benchmarking efforts.
* Further clarify and contextualize comments about gender. For example, why the focus on beards and distinct brow ridges in particular (L124-125)? The Ethical Considerations discussed in the paper are not especially actionable - perhaps there could be discussion about what contexts this dataset should be used in and when it may be misleading/insufficient.
* Explore fewer or more informative acronyms and include additional details, especially regarding dataset construction and experimental results (e.g. including the accuracy and DEO metrics on the counterfactual versions of the dataset as well).

**Relation To Prior Work:**

Please contextualize the contributions of the new dataset over prior work (https://ojs.aaai.org/index.php/AAAI/article/view/29333), as suggested in "Opportunities for Improvement"

**Summary And Contributions:**

This work introduces counterfactual versions of two existing datasets, CelebA and LFW, that are created using in-painting models. These datasets are also annotated and filtered according to the correctness of the attribute-related in-painting techniques and original target variable. The authors perform experimentation of classifiers trained with counterfactual interventions and propose a theoretical framework to explain disparities between counterfactual fairness and group fairness. They include additional empirical experimentation of these findings with a binarized version of the CIFAR-10 dataset.

---

> ### Author Rebuttal · Authors · 2024-08-14
>
> We thank the reviewer for taking the time to provide such a thorough review. We appreciate for recognizing our key contributions, e.g., the construction of counterfactual version of existing datasets and experimental and theoretical results for the relationships among the fairness notions. Your comments on the weaknesses and suggestions for improvement are also helpful to improve the paper. Following are our replies to all your comments. Please note that we will reply to "concerns" and the corresponding "opportunities for improvement" together.
> > Relatively little discussion of details about the new dataset
>
> The composition and creation process of the new datasets, CelebA-CF and LFW-CF, are detailed in Section H of the appendix. They are based on the real facial image datasets, CelebA and LFW. Both include labeling for 40 attributes and there exists an imbalance between these attributes. For example, in CelebA, most images with the blond hair attribute are female, and most males do not have blond hair, leading to bias between these two attributes. Similarly, in the new datasets we introduced, each facial attribute has a different distribution. We present the skewness values between the sensitive attribute and other non-sensitive attributes in CelebA-CF in the table below. Since the new datasets also include labeling for 40 attributes per image, we can leverage this information for various analyses.
> | Attribute | Hair_Length | Bangs | Wearing_Hat | Brown_Hair | Pale_Skin | Big_Lips | Mouth_Slightly_Open | Smiling | Wavy_Hair |
> |-|-|-|-|-|-|-|-|-|-|
> | Skewness | 0.798 | 0.907  | 0.974 | 0.820 | 0.980 | 0.815 | 0.265  | 0.335 | 0.833 |
>
> Additionally, we displayed the group-specific failure rate identified through the annotation process in the table below.
> ||CelebA-CF|LFW-CF|
> |-|-|-|
> |Female|0.57|0.70|
> |Male|0.80|0.69|
>
> > Contextualization with the existing work & Details about how the theoretical and empirical analyses may lend to broader benchmarking efforts
>
> First, we would like to emphasize that our counterfactual (CTF) datasets were created specifically to allow the measurement of counterfactual fairness (CF) alongside group fairness (GF) in the existing real-world datasets. Traditional image benchmark datasets used for fairness research lack CTF samples, making it impossible to measure CF and, consequently, to study the relationship between GF and CF. By editing these existing datasets to create counterfactual samples, we have enabled simultaneous benchmarking of both CF and GF. This is a significant distinction between our work and the study you mentioned [1]. Because [1] edits synthetically generated images to create CTF samples, it does not allow for measuring GF metrics based on real-world data. We will add this discussion in our final version of the paper.
>
> Using our benchmark datasets that enable a comprehensive study of the relationship between CF and GF, our observations demonstrated that GF-aware or CF-aware methods can fail to achieve these two fairness notions simultaneously, which contrasts with findings from previous research. Our theoretical results, along with experimental analyses using the CIFAR-10 dataset and the RFP on the auxiliary attribute G, were conducted to provide deeper insights into these benchmarking outcomes -- specifically, these all results consistently reveal that when the correlated attribute G exists, the equivalence between CF and GF may not hold. Therefore, this also aligns well with the theme of this "benchmarking" track.
> > Further clarify and contextualize comments about gender &  actionable ethical considerations
>
> We acknowledge that the counterfactual samples created using traits for "gender" could be subjective. Therefore, as suggested by ethical reviewer YaBR, we will revise our paper to use "sex" as the sensitive attribute instead of "gender" to avoid potential ethical issues. This change means that the attribute will refer to biologically distinguishable facial features such as facial hair and brow ridge, rather than "gender appearance."
>
> We will list these biological traits and update the instructions for filtering counterfactual samples accordingly, and we will regenerate our dataset. This revision aims to address the concerns related to the subjectivity that arises from using "gender appearance." (For more details on the dataset reconstruction, please refer to our response to YaBR.)
> > Explore fewer or more informative acronyms and include additional details
>
> Thank you for suggesting two ways of improving the readability and informativeness of our paper. We will ensure that the acronyms would be more informative in the final version, like using DCF and DGF instead of CD and DEO regarding the fairness metrics, and DCF_G instead of RFP regarding the metric of robustness to G.
>
> Regarding the additional experiments, we report the accuracy (acc-CTF) and DEO (DEO-CTF) for CKD and several baseline methods on counterfactual samples in CelebA-CF.
> ||Acc|DEO|
> |-|-|-|
> |Scratch|77.22|15.92|
> |SS|81.43|28.74|
> |RW|79.75|23.61|
> |LBC|78.06|33.95|
> |CP|75.21|62.82|
> |CKD|90.08|21.86|
>
> Following are our additional observations from the table. First, we observe that acc-CTF of Scratch significantly drops compared to the original dataset, indicating that many images originally classified correctly are now misclassified, which is consistent with the result of high CD in our paper. Second, GF-tailored methods like SS, RW, and LBC exhibit high DEO-CTF, unlike DEO on the original dataset. Based on our result in the manuscript showing CD values of these methods, this implies unbalanced prediction flipping between groups. Lastly, CKD achieves high acc-CTF compared to other methods, which is also consistent with low CD and high accuracy for the original test dataset shown in our manuscript. We will include these results in the final version of our paper.
>
> [1] Lui et al., Leveraging Diffusion Perturbations for Measuring Fairness in Computer Vision, AAAI, 2023.

---

> > ### Comment · Reviewer_sLrQ · 2024-08-23
> >
> > Thank you to the authors for their thoughtful consideration of my review. In particular, I appreciate the re-consideration of the wording around the visual attributes being perturbed in the dataset as well as the comparison among prior work and additional experimental details.

---

> > > ### Author Response · Authors · 2024-08-26
> > >
> > > Dear the reviewer sLrQ,
> > >
> > > We thank you for reviewing our paper and taking the time to read and consider our rebuttal. We are also glad that we were able to address the concerns you raised. If there are no further issues, would you consider raising the score for our paper?

---

> > > ### Author Response · Authors · 2024-08-29
> > > **Dataset reconstruction and re-evaluation results are uploaded**
> > >
> > > Dear Reviewer sLrQ,
> > >
> > > Thanks for your professional review again.
> > > We just uploaded the dataset reconstruction and re-evaluation results.
> > >
> > > Please check our new rebuttal. If you have any remaining concerns, please let us know.
> > > If our new rebuttal addresses your concerns, please consider raising your rating.
> > >
> > > Best,
> > >
> > > Authors

---

> > > > ### Author Response · Authors · 2024-08-31
> > > >
> > > > Dear Reviewer sLrQ,
> > > >
> > > > As the discussion period is coming to an end, we wanted to kindly ask if you might consider raising the score for our submission, provided there are no further concerns. If you have any additional questions or feedback, we would be more happy to address them.
> > > >
> > > > Thank you again for your time and thoughtful review.

---

### Official Review · Reviewer_dGaB · 2024-07-22
**Counterfactual dataset constructed needs more clarification**

**Rating:** 5
**Confidence:** 3
**Correctness:** Yes

**Review:**

Overall, I thought that the paper presented several interesting ideas, however, the main weakness (see W1 below) is concerning, hence my rating. I'm open to raising my rating if this is addressed sufficiently during the rebuttal process.

**Strengths:**

**S1**. The authors show, both theoretically and empirically, that models that improve counterfactual fairness may harm group fairness criteria.

**S2**. The authors propose a new method that optimizes for both group fairness and counterfactual fairness. The proposed method improves along both axes (CF and GF) unlike several prior methods.

**Additional Feedback:**

- L185: Theoritical analysis -> Theoretical analysis
- Fig. 2: Description of the nodes ($Y, A, G, I_k$) would help
- Tab. 5: A description of the bolded and highlighted rows would make the table more easy to read

**Clarity:**

- While the paper is mostly well written, the number of acronyms used made the paper hard to read .

**Documentation:**

Yes

**Limitations:**

Yes

**Opportunities For Improvement:**

**W1**. A major concern I have with this work is in the creation of the counterfactual dataset. Looking at the user interface (Appendix Fig. C.2), an image was considered incorrect if there were significant alterations in the persons' identity (e.g, hair style, skin color). However, it's unclear what alterations are considered significant - it seems like this controls for hair length / style, however, does not control for other variables like "facial hair", or "heavy makeup" (both of which seem to change in the example images provided). I agree with the authors that this is a key issue in constructing counterfactual datasets with images, since sensitive attributes like gender expression are heavily correlated with lots of attributes; but, it's unclear that the constructed dataset removes this correlation.

**W2**. Relatedly, I think it would be interesting to measure RFP (in Tab. 3) with respect to these other attributes that are also heavily correlated with gender expression.

**Relation To Prior Work:**

Yes

**Summary And Contributions:**

The authors show that image classification methods that optimize for counterfactual fairness can have worse group fairness than a vanilla baseline. They show, both theoretically, and empirically, that this could be due to an attribute that is heavily correlated with the sensitive attribute. The authors also construct a synthetic dataset based on CIFAR10 that allows users to vary both the bias of the sensitive attribute and the correlation b/w the sensitive attribute and another attribute. Finally, they propose a method that can simultaneously improve both counterfactual fairness and group fairness metrics.

---

> ### Author Rebuttal · Authors · 2024-08-14
>
> We appreciate your effort to review our paper in detail and point out our contributions, e.g., for our theoretical/empirical results on the connection between group-/counterfatual-fairness, and for our proposed method. Following are our replies to your comments on weaknesses.
>
> > W1
>
> First, we would like to note that the goal of our generation process was to generate counterfactual (CTF) samples when it comes to the sensitive attribute of "gender". Namely, for each CTF sample, only the attributes **caused** by this sensitive attribute are changed, while the other attributes (e.g., those that are merely **correlated**) are unchanged. While it is challenging to accurately distinguish between correlation and causation for all attributes, however, we relied on the implicit assumptions made by multiple human annotators regarding the relationships between attributes. For example, in Figure 1 showing the examples of our CTF samples, it seemed that the annotators implicitly assumed that features like facial hair and heavy makeup are caused by the gender appearance attribute, while hair length and style are simply correlated with it. Although we cannot guarantee that this process for creating CTF samples completely eliminates all features correlated, not caused, by the sensitive attributes, we believe that the careful annotations by multiple individuals have largely addressed this concern.
>
> We also note that to address the issues raised by the ethical reviewers, we plan to re-create the dataset by explicitly specifying instructions for secondary sex characteristics during the filtering process (please refer to our reply for Ethical Reviewer YaBr). We believe this approach will more effectively resolve the concerns you highlighted. Specifically, any characteristics other than the specified secondary sex characteristics (e.g., hair length) will be considered as correlated features, and human annotators will be able to filter out images where these features have changed, based on the revised instructions.
>
> > W2
>
> We thank you for your suggestion of measuring RFP on other attributes, which are correlated with the sensitive attribute. While we provided a method for selecting those kinds of correlated attributes more systemically in Appendix E.3, we would like to note that it is time-consuming to accurately modify only a single facial attribute in face images. (For measuring RFP regarding hair length, we manually applied strokes to 160 images to change the hair length). Although we were unable to complete the RFP measurement for other features during the rebuttal period, we will make sure to include these results in the final version of our paper.

---

> > ### Comment · Reviewer_dGaB · 2024-08-23
> >
> > I appreciate the comments by the authors regarding the dataset construction. Something I'd like to note is that annotators might reflect similar biases to that present within the dataset (as a note, the "true label" of gender expression is also from a consensus among annotators), thus, the difference between what attributes are correlated with or caused by gender is a product of that bias. I would highly recommend that when re-creating the dataset, the specifications are constructed after talking to experts as well as people with a wide range of lived experiences from different cultures and ethnicities, centering trans, non-binary, and/or gender non-conforming individuals.
> >
> > While I think that this work is promising, I do not believe that it is ready for publication at this stage.

---

> > > ### Author Response · Authors · 2024-08-29
> > > **Dataset reconstruction and re-evaluation results are uploaded**
> > >
> > > Dear the reviewer dGaB,
> > >
> > > Thanks for your professional review again. We just uploaded the dataset reconstruction and re-evaluation results.
> > >
> > > Please check our new rebuttal. If you have any remaining concerns, please let us know. If our new rebuttal addresses your concerns, please consider raising your rating.
> > >
> > > If you think our paper is not ready for publication, we would appreciate knowing why.
> > >
> > > Best,
> > >
> > > Authors

---

> > > > ### Comment · Reviewer_dGaB · 2024-08-29
> > > >
> > > > Thanks, I went through the newly constructed dataset and results shared. I do agree that this reduces the concerns. I have a couple of questions and suggestions to further improve the paper:
> > > >
> > > > - What do you mean by "considerations to makeup"? Looking at the pdf, it appears as though "makeup" is used as a secondary sex characteristic?
> > > >
> > > > - Similarly, it would help to include a discussion of why these characteristics were chosen, and what information was given to annotators about the characteristics (as a note, I'm not sure I understand what skin texture refers to) What if some characteristics are conflicting?
> > > >
> > > > - Finally, consider adding a discussion on how these choices might affect gender minorities, especially since the dataset is being released.
> > > >
> > > > I have increased my initial score.

---

> > > > > ### Author Rebuttal · Authors · 2024-08-29
> > > > >
> > > > > Thank you for taking the time to read our rebuttal and for your positive feedback.
> > > > >
> > > > > We are aware that makeup is not typically classified as a secondary sex characteristic. However, in the CelebA and LFW datasets we use, most feminine-like images include makeup (especially since many female celebrities in the CelebA dataset appear to be wearing makeup) and the IP2P model is biased towards removing makeup even when altering masculine features in these feminine images. Therefore, to prevent most images from being filtered out solely due to changes in makeup, we included makeup in the instructions, even though it is not a sex characteristic. We will clearly state in the paper that the CTF samples we created reflect sex characteristics, including makeup, and we will provide a detailed explanation of our reasons for including makeup. Importantly, we believe that as implicit bias toward gender or sex by human annotators is not used anymore by clearly specifying which facial attributes were considered, incorporating makeup into the instructions for our counterfactual datasets does not pose ethical issues.
> > > > >
> > > > > The facial features we used were selected by experts in fairness research. We initially identified 20 facial attributes using GPT-4o, then had three experts choose some facial attributes that best represent sex characteristics. To assist in this selection, we showed them samples of male and female images from existing benchmarks like CelebA and LFW, helping them determine which characteristics  represent biological males or females. Ultimately, we selected 9 attributes based on majority voting. We also provided annotators with a guideline that included instructions on how to interpret the selected facial attributes. Specifially, the guideline illustrated examples of counterfactual samples, showing cases where specific facial attributes were altered or remained unchanged. For your reference, skin texture refers to the roughness or smoothness of the skin on areas like the cheeks and forehead.
> > > > >
> > > > > Could you clarify what you mean about conflicting facial attributes? Since each facial attribute refers to a different part of the face, we believe there is little chance of these features conflicting with each other.
> > > > >
> > > > > Thank you for your valuable suggestion. Since the newly created dataset uses biological sex rather than social gender as the sensitive attribute, it is important for practitioners to be aware of this distinction when using our data. We believe that by ensuring a clear understanding of the facial attributes represented by the sensitive attributes in our dataset, we can significantly reduce any negative impacts on gender minorities. We will include this discussion in our paper.

---

> ### Author Rebuttal · Authors · 2024-08-26
>
> We fully agree with the reviewer’s comment. Labeling only with the instruction about whether gender has changed can introduce biases from the labelers’ own perceptions of gender expression into the dataset. However, we would like to emphasize that by following the professional advice of ethical reviewer YaBR and conducting additional survey from some experts specializing in algorithmic fairness, this issue can be significantly resolved.
>
> As mentioned in the previous rebuttal, we plan to redefine our sensitive attribute by replacing the term “gender” with “secondary sex characteristics” and update the instruction for filtering the counterfactual samples accordingly. To be more specific for filtering counterfactual samples in terms of our new sensitive attribute, we conducted a survey to seek expert guidance on identifying secondary sex characteristics. Initially, we selected 20 facial attributes using GPT-4o, then narrowed it down to 9 key attributes through a survey of fellow researchers specializing in algorithmic fairness: facial hair, Adam's apple, skin texture, jawline, chin shape, brow ridge, cheekbone prominence, lip fullness, and hairline. We incorporated them into the instruction that are used for filtering the counterfactual samples. This approach clarifies that these biological attributes have a causal relationship with sex, while non-biological attributes like hair length are merely correlated.
>
> As a result, our newly created dataset will no longer be influenced by labelers’ biases regarding gender expression. We are currently in the process of regenerating the data, and we will upload the new dataset along with the results for counterfactual fairness measured on this dataset within the discussion period.

---

### Official Review · Reviewer_pwn9 · 2024-07-25
**Review of "Do Counterfactually Fair Image Classifiers Satisfy Group Fairness? – A Theoretical and Empirical Study"**

**Rating:** 9
**Confidence:** 3

**Review:**

The paper presents a significant step forward in understanding the relationship between CF and GF in the context of image classification. Creating counterfactual samples in images, where typical sensitive attributes show complex traits in the image, is very challenging. The authors overcome this gap by leveraging state-of-the-art image editing methods to introduce two new datasets, CelebA-CF and LFW-CF, containing curated counterfactual samples that allow to study the relationship of CF and GF in this type of task. Beyond the empirical analyses, the theoretical analysis provides valuable insights into why CF does not imply GF in image classifiers. This motivated the proposal a new baseline method named Counterfactual Knowledge Distillation, showing potential in mitigating this issue by reducing reliance on latent attributes correlated with the sensitive attribute.

Quality: This is a very high-quality research, with a well-structured methodology and thorough experiments (including several methods, including CF-aware, GF-aware and CF- and GF-aware training methods), and including a sound theoretical analysis. The datasets created are valuable for the community, and the insights are well discussed and grounded.

Clarity: Overall, the paper is well-written and clear. However, the explanation of the human annotation process can be improved for better clarity (more details below).

Originality: This work is original in more than one dimension. Not only the proposed datasets are novel, but they open the door for much needed research on uniting CF and GF perspectives on model bias, in this case for image classification. The CKD method is also original, stemming from a sound theoretical analyses, and with promising empirical results.

Significance: This work has the potential to have a high impact in the FairML community, allowing further and more in-depth research into the relationship of CF and GF in image classification tasks. The benchmark is very well described and will be made available in full.

**Strengths:**

* Creation of high-quality (through curation) counterfactual image datasets, CelebA-CF and LFW-CF (and an additional CIFAR10-B), enabling evaluation of CF in image classification.
* Thorough theoretical and empirical analysis illustrating and justifying the disparity between CF and GF in image classifiers.
* Introduction of CKD, a promising method to reduce reliance on latent attributes correlated with sensitive attributes, improving both CF and GF.
* Comprehensive ablation studies and clear presentation of insights on CKD.

**Additional Feedback:**

N/A

**Clarity:**

The paper is generally well written, but improvements can be made in:
* Explaining the human annotation process more clearly: the two annotation stages are described in a way that the main criteria seem similar: "such as whether the gender change was accurate and whether other facial characteristics remained consistent with the original image" versus "Those annotators evaluate only the images that remained after the filtering, based on two criteria: (1) whether the sensitive at tribute was correctly changed and (2) whether the other non-sensitive attributes were preserved." After that, we can understand some differences but this should be made clearer.
* Discussing the impact of using IP2P, especially for augmentation where no filtering is applied.

**Correctness:**

The claims made in the submission are supported by the empirical results and theoretical analysis. The datasets are constructed in a sound way, and the evaluation methods and experiment design seem appropriate and performed correctly.

**Documentation:**

Between the main text body and supplementary materials, the datasets and benchmark are documented in sufficient detail. They will be made avaialble in a Github repository, and maintained by the authors. Data generation, organization is sufficiently described as well.

**Ethics:**

I cannot find significant ethical concerns with the submission besides the discussion posed by the authors regarding specific sensitive features like the binarization of gender and its analysis and manipulation in image classification. As an improvement point, the potential biases introduced by techniques like IP2P and the chosen human curators should be more clearly discussed.

**Limitations:**

The authors have addressed some limitations, such as the binary simplification of gender and the reliance on IP2P for data generation and human curation for filtering. However, further discussion on the potential biases introduced by non-curated IP2P on augmentation is beneficial. The authors could also consider adding more practical considerations in manipulating sensitive attributes in images given this is still a fairly complex pipeline, to support further research.

**Opportunities For Improvement:**

* The description of the two stages of human annotation is somewhat confusing. Both stages use similar criteria for evaluation, but the differences are not clearly explained. Clarity on the distinct targets and purpose of these stages is needed.
* The use of IP2P for CF-aware training and the amount of filtering shown for testing suggest potential negative impacts or bias, which are not thoroughly discussed. The authors should address the implications of using non-curated IP2P (versus the filtering on testing) on the overall findings.
* The authors could consider related work by Carreiro and Pinto et al., which introduced a counterfactual confusion matrix from which several metrics can be computed. Such an analysis could support a richer discussion beyond the Counterfactual Disparity (CD) metric used in this work.

**Relation To Prior Work:**

The literature review effectively contextualizes the research within the field. The authors discuss how this work differs from previous contributions by highlighting the unique aspects of CF and GF in image classification, also compared to previous work and conclusions on tabular data. As a suggestion, the authors could consider incorporating an analysis based on the counterfactual confusion matrix by Carreiro and Pinto et al. (https://dmlr.ai/assets/accepted-papers/85/CameraReady/ICML_2023_DMLR_Workshop_Counterfactual_Confusion_Matrix.pdf) which could provide a richer discussion and more comprehensive evaluation of CF and GF. That paper also seems to argue that CF does not imply GF, in tabular datasets.

**Summary And Contributions:**

This work investigates the relationship between counterfactual fairness (CF) and group fairness (GF) in image classification, an under-explored but relevant area of research. The authors introduce new datasets, CelebA-CF and LFW-CF (and a modification to CIFAR-10 to a binary task), to evaluate CF by generating high-quality counterfactual images through curated image editing. The main findings show that CF does not necessarily imply GF in image classification, in contrast to most previous studies on tabular data. The authors further propose a method named Counterfactual Knowledge Distillation (CKD) to address this issue, by reducing the correlation of latent variables with the protected attribute (and its relationship with the target). The main contributions are:

* Creation of CelebA-CF and LFW-CF (and CIFAR-10B) datasets for evaluating CF in image classification.
* Theoretical analysis explaining the discrepancy between CF and GF due to latent attributes correlated with the sensitive attribute.
* Introduction of a new baseline method (CKD) to achieve both CF and GF by reducing reliance on the latent attributes.

---

> ### Author Rebuttal · Authors · 2024-08-14
>
> Thank you very much for your detailed and thoughtful review of our work. We are pleased that you found our contributions valuable, including the creation of high-quality image datasets, the theoretical and empirical analyses of the relationship between counterfactual fairness (CF) and group fairness (GF) in image classification, and the development of a new method achieving the two fairness notions. We also value your constructive feedback on areas for improvement. Following are our replies to your comments.
>
> **Opportunities for improvement**
> > More clarity on the human annotation process
>
> We apologize for the unclear explanation of the differences between the two stages of human annotation. Both processes are similar in that they involve checking for changes in specific attributes. However, the first stage is used to create the datasets, while the second stage is used to validate the quality of these datasets. Specifically, in the second stage, we selectively focus on a few non-sensitive attributes for more detailed verification of counterfactual (CTF) images.
> > The implication of using non-curated IP2P
>
> Thank you for suggesting a good discussion point for training with non-curated CTF datasets. As you pointed out, uncurated CTF datasets are imperfect. Specifically, some samples generated from the original images in our test datasets were filtered out because the images either showed minimal changes or had alterations that affected non-sensitive attributes including the target attributes. Consequently, the more such incomplete samples exist, the more they will negatively impact the performance of our method.
>
> To assess how sensitive CKD is to incomplete CTF training samples, we conducted additional experiments on CIFAR-10B by varying the ratio of incomplete CTF samples in the training set. For a given ratio, $\alpha$, we assumed that half of the incomplete samples are nearly unchanged, while the other half are samples where both the target and sensitive attributes are altered. We varied $\alpha$ from 20% to 60% in 10% increments and reported the accuracy, CD, and DEO of CKD in the table below. The results indicate that CKD significantly improves both CD and DEO compared to Scratch, even for high $\alpha$s. Although this phenomenon has not been fully explained, we hypothesize that the robustness can be attributed to the distillation process, as empirically demonstrated in [1].
> ||Acc|CD|DEO|
> |-|-|-|-|
> |Scratch|78.01|17.90|27.46|
> |CKD|78.49|2.85|7.30|
> |CKD (20%)|79.82|2.77|11.41|
> |CKD (30%)|79.76|2.88|12.78|
> |CKD (40%)|79.70|2.94|12.88|
> |CKD (50%)|79.72|2.85|14.10|
> |CKD (60%)|79.61|3.01|14.94|
> > Analysis of related work by Carreiro and Pinto et al.
>
> We first note that Counterfactual Disparity (CD) we used is the same metric as Switch Rate (SR) proposed by Carreiro and Pinto et al. As the reviewer mentioned, utilizing the other metrics proposed in that paper can provide a more comprehensive interpretation of our results. For example, we computed P2NR (another metric proposed by the paper) on CelebA and obtained values of 0.036, 0.165, and 0.339 for Scratch, CP, and CKD, respectively. These results indicate that CKD achieves low CD with a balanced rate of misclassification across the labels of original samples. We will address results for other metrics as well in the final version of our paper.
>
> Additionally, we would like to emphasize that Carreiro and Pinto et al. focused on scenarios where GF does not imply CF in their experiments—highlighting cases where the faithfulness assumption, which can be overly stringent, does not hold (see line 323 in their paper). However, our work primarily explores the converse: whether CF can imply GF depending on the presence of $G$, independent of the faithfulness assumption.
>
> **Limitations**
>
> We note that the first limitation you mentioned is addressed in the second response of the "Opportunities for Improvement" section.
> > More practical considerations in manipulating sensitive attributes
>
> We respectfully disagree with your comment on the limitation of the complexity of generating counterfactual samples. IP2P is pretty practical, as it can edit images using simple text instructions only. Furthermore, the labeling and verification steps are inevitable to ensure the quality of benchmark datasets. Nevertheless, we hope that the process of generating counterfactual samples could be more efficient if more sophisticated editing techniques were developed.
>
> **Clarity**
>
> The two issues mentioned can be found in the first two responses under the "Opportunities for improvement" section above.
>
> **Relation to Prior Work**
>
> Please see the third response in the "Opportunities for Improvement" section.
>
> **Ethics**
> > Potential biases introduced by techniques like IP2P and the chosen human curators
>
> As ethical reviewer ae5K pointed out, when creating CTF samples for gender, which is our sensitive attribute, we did not provide specific visual characteristics. As a result, the inherent assumptions about gender and certain facial attributes held by the IP2P model and human annotators were embedded in our CTF samples. For instance, as shown in the filtered CTF and original image pairs in Figure 1, these assumptions suggest that features like facial hair and brow ridge are causally related to gender, while hair length is merely correlated.
> Using such assumptions, as ethical reviewer YaBR mentioned, can lead to some ethical concerns by categorizing a social identity like gender based on visual traits. Therefore, following YaBR’s suggestion, we have decided to replace the term "gender" with "sex" for our sensitive attribute and revise our paper to focus on secondary sex characteristics. We will also regenerate the dataset accordingly with the renewed sensitive attribute. (For detailed plans, please refer to our response to YaBR.)
>
> [1] Goldblum et al., Adversarially robust distillation, AAAI, 2020.

---

> > ### Comment · Reviewer_pwn9 · 2024-08-29
> >
> > Although I can't see the revised version of the paper, I appreciate the authors effort in replying to my comments and concerns, including a new experiment on incomplete CTF training samples, new counterfactual metrics and corresponding discussion, and a major effort in re-building the datasets to address ethical concerns.
> >
> > A minor comment regards my previous comment on "More practical considerations in manipulating sensitive attributes". What I meant was for the authors to provide hints on the process of manipulating sensitive attributes in images that could guide future work: challenges, specific limitations. Some were raised by other reviewers, like correlated vs caused by, conflicting features, etc.
> >
> > Thank you,

---

> > > ### Author Rebuttal · Authors · 2024-08-31
> > >
> > > Thank you for acknowledging our efforts to address all your comments, and we appreciate your suggestions for improving our paper. While we haven’t submitted the revised version of our paper yet, we make sure that everything discussed during the rebuttal period, including the following point, will be incorporated into the final manuscript.
> > >
> > > Regarding “more practical considerations in manipulating sensitive attributes,” we can further discuss the following aspects. First, as noted by other reviewers, using slightly abstract sensitive attributes like “gender” and only mentioning it in the instructions may lead to biased labeling or an unclear causal structure. To mitigate this, we revised our data labeling guidelines and recreated the datasets. We strongly encourage practitioners to specify the exact attributes their sensitive attribute means or refers to, or to use more detailed causal graphs. Second, IP2P may sometimes fail to accurately edit images for certain attributes. Similar to our filtering process, labelers can refine edited images by removing only those that were poorly edited. However, if the IP2P model fails to edit most images, the editing performance can be improved by fine-tuning diffusion models or by adopting more time-consuming yet efficient editing methods like [1].
> > >
> > > [1] Kwon et al., Diffusion Models already have a Semantic Latent Space, ICLR, 2023.

---

### Author Rebuttal · Authors · 2024-08-14

### Response to ethics review by ethics reviewer ae5K

We acknowledge your comment regarding the implicit assumptions we utilized about the relationship between "gender" and facial attributes.

First, we want to clarify that, from an experimental standpoint, these assumptions do not present an issue. As long as there is consensus on this relationship (e.g., attribute A is caused by gender, while attribute B is merely correlated with it), our experimental and theoretical results, which demonstrate that CF cannot lead to GF due to attribute B, remain valid.

However, as the ethical reviewer YaBr pointed out, the term "gender" can include the meaning related to social identity, and using binary classifications of "gender" based on visual traits can risk excluding individuals who do not conform to these traits. Therefore, in line with YaBr's suggestion, we plan to revise our paper to adopt the terms "sex" and "sex-related secondary characteristics" instead of "gender" and "gender appearance", respectively, for defining our sensitive attribute. Additionally, we plan to revise the counterfactual sample filtering instructions for human annotators based on this new sensitive attribute and re-create the datasets accordingly until the discussion period (for more details on the revisions, please refer to our response to YaBr). By making these changes, we believe that we can more explicitly determine which visual traits should be considered when making counterfactual samples, therefore significantly mitigating the normative harms previously mentioned.

---

### Author Rebuttal · Authors · 2024-08-14

### Response to ethics review by ethics reviewer YaBr

We deeply acknowledge your ethical concerns regarding the classification of 'gender' and appreciate your thoughtful suggestion to address this issue. We have decided to adopt your recommendation and will replace the term 'gender' with 'sex' for our sensitive attribute throughout our paper. This change allows the sensitive attribute to more explicitly refer to sex-related secondary characteristics, such as 'facial hair' and 'brow ridge', rather than socially constructed concepts related to 'gender'. By using 'sex', which focuses more on the concepts of biological distinctions in facial images, we believe that many of the ethical and normative issues-such as the misclassification of 'gender' based on visible traits or the perpetuation of misconceptions about the concept of gender-can be significantly mitigated.

We note that even with this change in the term and concept of our sensitive attribute, the core contributions of our work remain intact. Specifically, the proposed pipeline for creating counterfactual samples, which involves using IP2P for image editing and human annotators for filtering, can be adapted by slightly modifying the instructions for annotators to focus on changes in sex-related secondary characteristics rather than gender. This means we can still create counterfactual benchmark datasets using the revised sensitive attribute (please refer to our plan for the re-labeling of our datasets below). Furthermore, our experimental and theoretical results, which demonstrate the potential inequivalence between CF and GF in image classification, remain valid with the sensitive attribute defined as 'sex', because facial attributes such as hair length, which are correlated with but not caused by 'sex' (i.e., the secondary sex characteristics), still exist in the images. Additionally, our proposed methodological mechanism for reducing reliance on such attributes to simultaneously achieve CF and GF, is also unaffected by whether the sensitive attribute is defined as 'sex' or 'gender'.

To create a counterfactual dataset that aligns better with the new sensitive attribute term, we plan to recreate the counterfactual datasets before the discussion period ends (by August 31). Specifically, for the filtering process of incorrect counterfactual samples, we will revise the instructions to explicitly mention sex characteristics instead of using the term 'gender appearance', and have human annotators re-label the generated images. For this, we compiled a list of 20 secondary sex characteristics using GPT-4o and manually selected the following attributes: facial hair, jawline, chin shape, brow ridge, forehead slope, cheekbone prominence, nose size, lip fullness, adam's apple, hairline, and skin texture. One notable issue is that most of the feminine-like images in the benchmark datasets we use (CelebA and LFW) include makeup (for instance, many female celebrities in the CelebA dataset appear to be wearing makeup) and the bias of IP2P eliminates makeup in female images when altering masculine characteristics. Thus, to prevent most of the generated samples from being filtered due to alteration of makeup, we will include the presence of makeup as a criterion in the filtering instructions. After recreating the counterfactual datasets, we will also re-evaluate CF for all models in our paper using the newly created dataset and report the results. By doing so, we aim to ensure that our research on benchmarking CF and GF could be both ethically more sound and free from potential concerns.

We would like to note that despite modifying the instructions to regenerate the dataset, we do not expect the newly generated data to differ significantly from the original dataset. This is because the revised instructions based on sex characteristics are simply more specific rather than conflicting with or diverging from the criteria we previously used. In a preliminary study where the authors applied the revised instructions to filter 200 pairs of original and counterfactual samples, only 9 additional pairs were filtered out compared to the original dataset. Although this filtering was conducted by just two annotators and may not be entirely precise, it still suggests that the regenerated dataset will not differ substantially from the original. Consequently, we anticipate that our experimental results on benchmarking CF will also remain largely unchanged.

We would appreciate for letting us know your thoughts on our proposed revision plan. Do you have any concerns or further suggestions regarding our plan? Please feel free to share your feedback.

---

### Author Response · Authors · 2024-08-22
**Follow-up on Rebuttal and feedback on dataset reconstruction**

Dear reviewers,

We thank the reviewers for taking the time to review our work. Could you please confirm if you have had a chance to look at our rebuttal? We’ve put in a lot of effort and are wondering if it might be possible to improve the scores. Also, we are planning to reconstruct our dataset to address the ethical issues posed by the ethical reviewers. Do you have any comments or feedback regarding this?

---

### Author Rebuttal · Authors · 2024-08-29

### Dataset reconstruction and re-evaluation results

Dear reviewers,

As mentioned in our rebuttal, we have recreated the dataset and re-evaluated counterfactual fairness (CF) for the models used in our paper in order to address the ethical concerns raised. Following the advice of ethical reviewer YaBr, we revised the term of the sensitive attribute from “gender” to “sex,” and ensured that it now refers to "secondary sex characteristics" rather than the appearance of “man” or “woman.”

Specifically, we extracted 20 facial attributes using Chat-GPT and then, with guidance from experts specialized in fairness, selected 9 key facial attributes (facial hair, Adam's apple, skin texture, jawline, chin shape, brow ridge, cheekbone prominence, lip fullness, and hairline). We updated the instructions for human annotators to filter out counterfactual samples based on these attributes (including considerations for makeup). Using these revised instructions, five labelers re-labeled the generated counterfactual samples. We filtered out samples receiving two or fewer votes.

As a result, from the original 720 and 632 CelebA and LFW counterfactual sample pairs, we retained 230 and 144 pairs, respectively. This is slightly fewer than the previous 237 and 193 pairs, with 181 and 115 pairs overlapping between the old and new filtered datasets.

We then re-calculated the counterfactual fairness metric, CD, for all models evaluated in the manuscript using the newly created counterfactual sample pairs. The table below compares the CD values obtained with the new dataset to those from the previous datasets for the models in Table 5 of the manuscript. As shown, while the CD values vary slightly, all the claims made in the paper remain intact. For instance, baselines for counterfactual fairness achieve lower CD values but fail to achieve group fairness, whereas our CKD method successfully achieves both group and counterfactual fairness.

We provide access to our newly created dataset, i.e., CelebA-CF and LFW-CF, through the following link: CelebA-CF (https://figshare.com/s/62b6f7f69d0eab9c3c71), LFW-CF (https://figshare.com/s/39f2daac58148e10e5fe). Additionally, the snapshot of the instructions used for data generation and all CD results recalculated with the new dataset are included in the attached PDF.

We believe that by following ethical reviewer YaBr’s advice and rebuilding the dataset, we have significantly addressed the ethical issues raised by the reviewers, such as the harms associated with labeling based on gender appearance. If there are any remaining concerns, please let us know. Thank you for your consideration.


| Method    | CelebA (and CelebA-CF) |               |       |       |     | LFW (and LFW-CF) |               |       |       |
| --------- | ---------------------- | ------------- | ----- | ----- | --- | ---------------- | ------------- | ----- | ----- |
|           | Acc ↑                  | CD (before) ↓ | CD ↓  | DEO ↓ |     | Acc ↑            | CD (before) ↓ | CD ↓  | DEO ↓ |
| Scratch   | 95.53                  | 10.21         | 10.36 | 47.10 |     | 90.85            | 17.81         | 18.06 | 7.66  |
| SS        | 95.44                  | 8.99          | 8.41  | 42.95 |     | 90.43            | 18.79         | 18.19 | 6.75  |
| RW        | 95.16                  | 5.80          | 5.50  | 24.21 |     | 90.87            | 17.33         | 18.68 | 6.92  |
| COV       | 94.42                  | 8.12          | 7.72  | 34.04 |     | 90.85            | 14.93         | 16.43 | 6.99  |
| MFD       | 94.37                  | 5.45          | 4.61  | 19.00 |     | 90.47            | 19.44         | 16.07 | 2.15  |
| LBC       | 94.92                  | 5.69          | 6.24  | 22.61 |     | 90.71            | 15.15         | 15.76 | 3.56  |
| SS+aug    | 95.17                  | 5.72          | 4.13  | 40.80 |     | 89.96            | 14.06         | 15.23 | 6.82  |
| RW+aug    | 95.13                  | 5.18          | 5.34  | 24.63 |     | 90.76            | 17.68         | 18.63 | 6.71  |
| COV+aug   | 94.08                  | 8.07          | 8.11  | 29.03 |     | 90.47            | 12.83         | 13.65 | 6.78  |
| MFD+aug   | 93.78                  | 4.16          | 3.87  | 14.36 |     | 89.90            | 20.20         | 19.36 | 2.47  |
| LBC+aug   | 94.39                  | 9.09          | 9.32  | 36.08 |     | 88.66            | 14.41         | 12.41 | 2.79  |
| CP        | 94.10                  | 2.16          | 2.53  | 51.01 |     | 89.77            | 7.75          | 9.2   | 8.74  |
| SS+CP     | 94.54                  | 2.16          | 2.40  | 37.97 |     | 88.70            | 5.16          | 6.13  | 4.26  |
| RW+CP     | 95.19                  | 5.00          | 4.67  | 25.56 |     | 90.87            | 14.76         | 15.24 | 6.16  |
| COV+CP    | 94.29                  | 5.89          | 5.36  | 51.63 |     | 91.23            | 11.03         | 11.91 | 6.52  |
| MFD+CP    | 93.81                  | 3.39          | 3.47  | 23.31 |     | 89.39            | 12.94         | 15.15 | 1.90  |
| LBC+CP    | 95.12                  | 4.76          | 4.72  | 22.78 |     | 89.92            | 7.98          | 8.33  | 3.02  |
| CKD (λ=0) | 94.12                  | 4.53          | 4.31  | 14.11 |     | 90.76            | 10.83         | 12.47 | 2.64  |
| CKD       | 93.08                  | 4.43          | 4.44  | 13.23 |     | 89.26            | 7.15          | 7.94  | 1.88  |

---

### Decision · Program_Chairs · 2024-09-26

**Decision:**

Accept (Poster)

**Comment:**

The paper analyses the relationship between counterfactual fairness and group fairness in image classification.
The authors contributed two new datasets and a modified CIFAR-10.
Counterfactual fairness is evaluated by generating high-quality counterfactual images through curated image editing.
An interesting insight/demonstration is that the counterfactual fairness does not imply group fairness.
The authors contributed Counterfactual Knowledge Distillation to address the identified issue explaining the phenomenon - the correlation of latent variables with the sensitive / protected attribute.
In their rebuttal, the authors provided convincing responses to the critique of the reviewers.